# SUBench: Benchmarking Spatial Understanding in Vision-Language Models

## Abstract

Recent advancements in vision-language models (VLMs) have shown remarkable success in general image-text retrieval. However, their ability to understand spatial relationships within images remains undertested. To address this gap, in this paper, we introduce **SUBench**, a large-scale dataset of more than 50k image-text pairs meticulously designed to evaluate a wide range of spatial relationships from real-world images. To curate the dataset, we designed an LLM-based framework to align human subjective descriptions with objective spatial relationships. In addition, unlike existing benchmarks, SUBench features a principled taxonomy of spatial concepts to ensure clarity and reduce ambiguity, alongside a scalable pipeline that systematically generates challenging hard negatives. Our experiments show that even state-of-the-art CLIP models struggle significantly on SUBench, revealing a critical blind spot in the spatial understanding capabilities of modern VLMs. Furthermore, we use the same approach to curate a training set and demonstrate that finetuning on this data not only improves performance significantly on SUBench but also enhances results on existing evaluation benchmarks. We will release the benchmark and believe SUBench will serve as a valuable resource to facilitate the development of more spatially-aware VLMs.

## 1 Introduction

Human cognition of the physical world is a synthesis of perceiving objective reality and forming subjective, embodied experiences within it. In contrast, contemporary machine intelligence—including Vision-Language Models (VLMs) (Radford et al., 2021; Maninis et al., 2025; Zhai et al., 2023; Tschannen et al., 2025) and Multi-Modal Large Language Models (MLLMs) (OpenAI et al., 2024; 2023; Deepmind et al., 2023; 2025; Xu et al., 2025)[1]—learns exclusively from curated data. These models process vast collections of images and text that represent the world objectively, but they lack any first-person, subjective perspective from which to ground their knowledge.

This fundamental distinction in their learning paradigms leads to a clear divergence in capabilities. VLMs (Radford et al., 2021; Tschannen et al., 2025) have demonstrated remarkable success in mining rich semantic information from visual data. However, this proficiency in the semantic domain does not naturally translate to an intuitive grasp of spatial relationships. Concepts such as "on", "behind" or "inside" are learned as statistical patterns in pixel arrangements rather than as lived, physical realities. Consequently, as shown in Figure 1, their understanding of where things are, and how they relate to each other in space, remains superficial compared to human intuition (Kamath et al., 2023; Khemlani et al., 2025; Cai et al., 2025). Compared to MLLMs (Deepmind et al., 2025; Xu et al., 2025) with advanced reasoning capabilities, VLMs function primarily as powerful feature extractors rather than as agents capable of complex spatial inference.

To effectively probe the spatial understanding deficiencies in VLMs, the image-text retrieval task serves as an ideal framework. Crucially, it enables the creation of hard negatives—pairs where entities in the text and image match semantically, but their spatial relationship is deliberately falsified. Such samples are essential for challenging models that rely on superficial semantic shortcuts rather than genuine spatial comprehension. Prior efforts to benchmark spatial understanding, however, have notable limitations. Human annotation methods, such as Visual Spatial Reasoning (Liu

---

[1]In this paper, VLMs refer to models such as CLIP, and MLLMs refer to models such as Gemini.

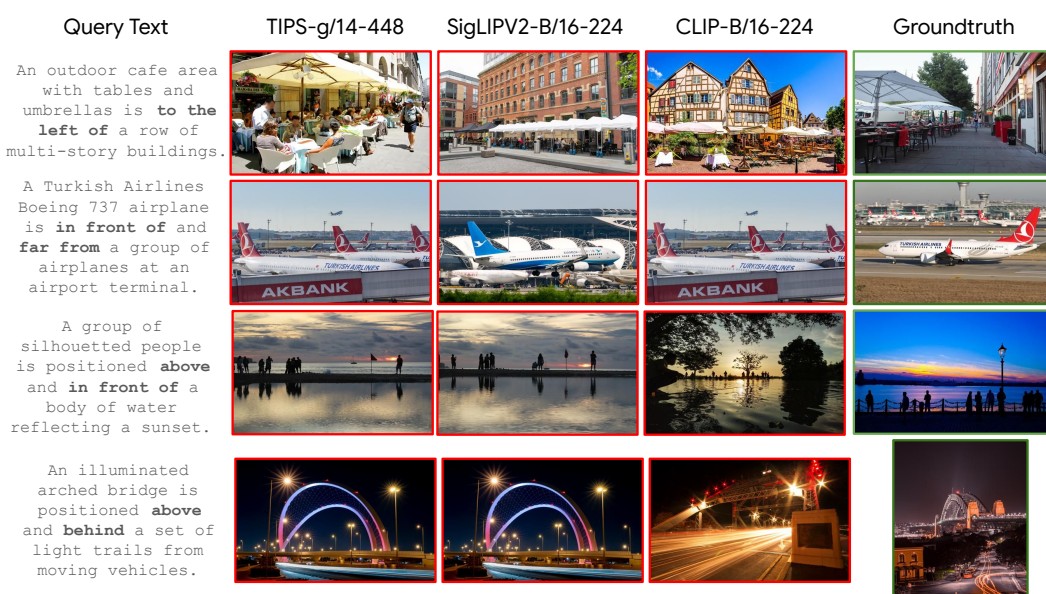

Figure 1: Top retrieval results from SUBench using three different modern VLMs. Spatial relationships are highlighted in bold in the query text, and the correct images (groundtruth) are highlighted with green boundary. For all four examples, the VLMs identifies the correct objects in the images but fails to understand the spatial relationships.

et al., 2023) and OmniSpatial (Jia et al., 2025), were constrained by limited scale and the inherent ambiguities of intuitive labeling. Other works (Chen et al., 2024a; Yang et al., 2025; Shiri et al., 2024) have leveraged 3D perception alongside predefined linguistic templates, which restricts the expressive freedom of natural language and makes the data unsuitable for retrieval tasks. Our approach is guided by the principle that spatial cognition is a synthesis of objective physical reality and subjective linguistic framing (linguistic relativity). Therefore, an ideal benchmark must align with human language patterns, employ a precisely defined relational vocabulary to eliminate ambiguity, and support large-scale data generation to enable cross-modal retrieval.

In this paper, we propose **SUBench**, which consists of 50k text-image pairs with explicitly mined hard negative samples, totaling more than 100k images. Our construction process is distinguished by a guideline-driven approach that uses Gemini 2.5 Pro's (Deepmind et al., 2025) Chain-of-Thought reasoning, constrained by principles designed to produce consistent, challenging, and human-centric spatial descriptions. A disjoint training split was also created using the same process, and we demonstrate that fine-tuning CLIP models on this data yields significant improvements on SUBench without compromising existing text-image retrieval performance.

The main contributions of this work are:

- We present **SUBench**, a high-quality dataset of 50k image-text pairs for the cross-modal retrieval task, designed to probe spatial understanding capabilities. Our benchmark, created using a guideline-driven approach, introduces both image and text hard negatives to evaluate VLMs.
- Our experiments reveal that existing VLMs, despite strong cross-modal retrieval capabilities on general benchmarks, struggle significantly on SUBench, which focuses on spatial relationships.
- We explore enhancement strategies for VLMs from both data and modeling perspectives, demonstrating a clear path toward improving spatial-aware cross-modal retrieval.

## 2 RELATED WORK

**Vision-Language Models.** Contrastive Language-Image Pre-training (CLIP (Radford et al., 2021)) has emerged as a dominant paradigm for learning powerful, multi-modal representations from web-scale data. By scaling the data (Schuhmann et al., 2022; Gadre et al., 2023; Li et al., 2024; Dong

et al., 2025; Wang et al., 2025) and model size (Radford et al., 2021; Zhai et al., 2023; Sun et al., 2023; Chen et al., 2024b; Tschannen et al., 2025; Maninis et al., 2025; Bolya et al., 2025; Chuang et al., 2025; Siméoni et al., 2025), performance has been shown to consistently improve with larger models and vast datasets containing billions of image-text pairs. The primary benchmark for measuring these improvements is image-text retrieval (Chen et al., 2015; Young et al., 2014; Onoe et al., 2024), which assesses a model's ability to match images to corresponding textual descriptions and vice versa within a shared embedding space. At the same time, the task of image-text retrieval is not merely an academic benchmark but also a foundational technology for a wide range of practical applications, such as large-scale image search engines where users can find relevant visuals using natural language queries (Tang et al., 2025; Karthik et al., 2024). Therefore, improvements in retrieval performance directly translate to more intuitive and powerful tools for information access.

**Spatial Understanding and Reasoning Benchmarks.** Pioneering works in creating spatial understanding benchmarks used structured representations such as scene graphs (Johnson et al., 2015; Ost et al., 2021), which explicitly model objects and their relationships. However, generating comprehensive scene graphs from complex images is difficult, challenging to scale, and not well-aligned with human language. With the advent of powerful Vision-Language Models (VLMs) (Radford et al., 2021) and Multimodal Large Language Models (MLLMs) (OpenAI et al., 2024), it is now possible to directly translate visual scenes into natural language. While this approach is more scalable and intuitive, it introduces the critical challenge of ensuring that the generated language accurately preserves the spatial relationships depicted in the image (Cai et al., 2025; Jia et al., 2025). Consequently, there is a need for benchmarks that can rigorously evaluate a model's ability to comprehend and articulate these fine-grained spatial details. To fill this gap, some works (Liu et al., 2023; Kamath et al., 2023; Jia et al., 2025) rely on human annotators to construct Visual Question Answering (VQA) datasets. While valuable, these benchmarks are often limited in scale and are constrained to the VQA format, which is not well-suited for evaluating the retrieval capabilities of VLMs. Furthermore, they may struggle to tackle the ambiguities inherent in natural language. Another line of research (Chen et al., 2024a; Cheng et al., 2024) uses 3D perception models and pre-defined templates to synthetically generate data. This approach also primarily targets VQA and faces difficulties in creating natural language descriptions, particularly when grouping same-category objects. This may result in captions that do not align well with human intuition and perception. In this paper, we use a guideline-driven approach that leverages a Large Language Model's Chain-of-Thought (CoT) capability to generate text based on constrained principles, creating a scalable pipeline for producing diverse and accurate spatial captions. This method yields a large-scale benchmark rich in natural language, specifically tailored for the nuanced evaluation of VLMs rather than VQA.

**Synthetic Data Generation.** The high cost and effort of manual annotation have made synthetic data a crucial strategy for training robust machine learning models. The emergence of LLMs has been a game-changer in this domain. In natural language processing, LLMs are routinely used for data augmentation (Borisov et al., 2023) and for generating new examples for tasks such as text classification (Li et al., 2023) and question-answering (Puri et al., 2020). This paradigm has recently extended into the multimodal domain, where LLMs are leveraged to generate rich textual descriptions and pseudo-annotations for images (Chen et al., 2022; Garg et al., 2024; Deng et al., 2025; Li et al., 2025). Our work builds on this trend but focuses specifically on the challenge of spatial understanding. We develop a framework that utilizes Gemini 2.5 Pro (Deepmind et al., 2025) both as a general generator and as a reasoning verifier, constrained by principled rules to mine complex, nuanced spatial relationships and, importantly, to curate "hard negative" samples. Using this approach, we also create a training dataset designed to improve the spatial awareness of VLMs. This targeted data generation framework represents a novel application of LLMs that goes beyond simple data augmentation, providing a powerful training signal that encourages VLMs to develop a more precise and robust understanding of spatial concepts.

## 3 Method

### 3.1 A Guideline-Driven Benchmark

Our benchmark is designed for image-to-text retrieval tasks, which demands fine-grained visual-textual alignment. To avoid vague descriptions which might provide weak learning and evaluation

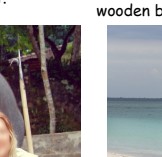
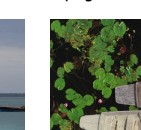

A woman with blonde hair is in front of and to the right of an Asian elephant.

A group of three people riding horses is in front of and distant from a group of traditional wooden boats.

A group of people wearing straw hats is lying inside two wooden boats.

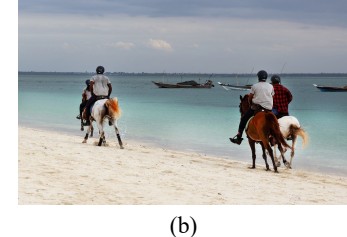
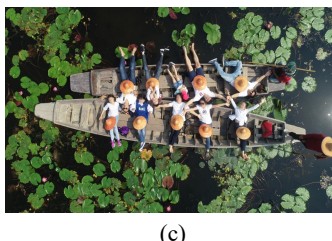

(a)                                    (b)                                    (c)

Figure 2: **Guidelines of SUBench.** Our benchmark tests: (1) diverse spatial relations, including (a) directional, (b) distance-based, and (c) topological; (2) reasoning over groups of objects (b, c); and (3) perspective-dependent relations within an egocentric frame of reference (a).

signals, we use a guideline-driven process to generate precise and unambiguous image-text pairs. This process is centered on the following principles:

**Egocentric Frame of Reference.** All directional relationships (e.g., left, above) are defined strictly from the viewer's perspective to eliminate ambiguity. Other frames of reference, such as an object's intrinsic orientation (e.g., the elephant's left) or an absolute direction (e.g., north), are unreliable for image-to-text retrieval because a user searching for an image lacks this information. For example, in Figure 2(a), the woman is described as "to the right of" the elephant, which is true from the viewer's (egocentric) perspective. From the elephant's (intrinsic) perspective, she would be on its left. By enforcing an egocentric frame, we ensure that our textual descriptions align with the most intuitive and consistent viewpoint, creating a more robust and unambiguous benchmark.

**Standardized Labels and Descriptive Captions.** To represent scene information, we adopt a dual approach: a standardized label for computational consistency and a descriptive caption for semantic richness. The label uses a controlled vocabulary, mapping synonyms like "on", "atop", and "upon" to a single, unambiguous term. This ensures clean, reliable data for classification and evaluation. Note that synonym mapping is applied only to the label space and the caption preserves the original natural language description, retaining crucial context that standardization would lose. This system combines machine-readable consistency with rich, human-level details.

**Object Grouping.** We manage visual complexity by treating collections of similar objects as a single conceptual group. This approach aligns with how humans perceive the "gist" of a scene and avoids a combinatorial explosion of redundant, verbose descriptions. For example, instead of describing each person and boat individually in Figure 2(b) and 2(c), we refer to them collectively (e.g., a group of people"). This allows us to capture the essential structure of the image rather than fine-grained details. We intentionally focus on the scene's primary relations for retrieval, rather than exhaustively enumerating intra-group relations, to align with human perception.

**The Salience Principle.** We describe only informative relationships, filtering out trivially true statements. Generic descriptions like "The sea is below the sky" are excluded because they describe default conditions and lack the discriminating power needed to identify a specific image within a large dataset. By focusing on salient and visually significant details, we capture the unique compositional essence of each scene. This ensures our dataset is both effective and discriminating.

## 3.2 BENCHMARK CONSTRUCTION PIPELINE.

Our benchmark construction process is a three-stage pipeline designed to produce a comprehensive and challenging dataset for spatial understanding, as illustrated in Figure 3. In the first stage, we generate the initial set of image-text pairs specifically focusing on spatial understanding task. This is achieved using a meticulously designed guideline (Sec. 3.1) to prompt a Large Language Model (LLM), ensuring high-quality and relevant captions. In the second stage, we create hard negatives by negating the textual descriptions, altering key spatial relationships to be incorrect. The third stage further increases the difficulty by introducing hard negative images, which are visually similar to the originals but invalidate the corresponding caption's spatial description. As a result, this pipeline

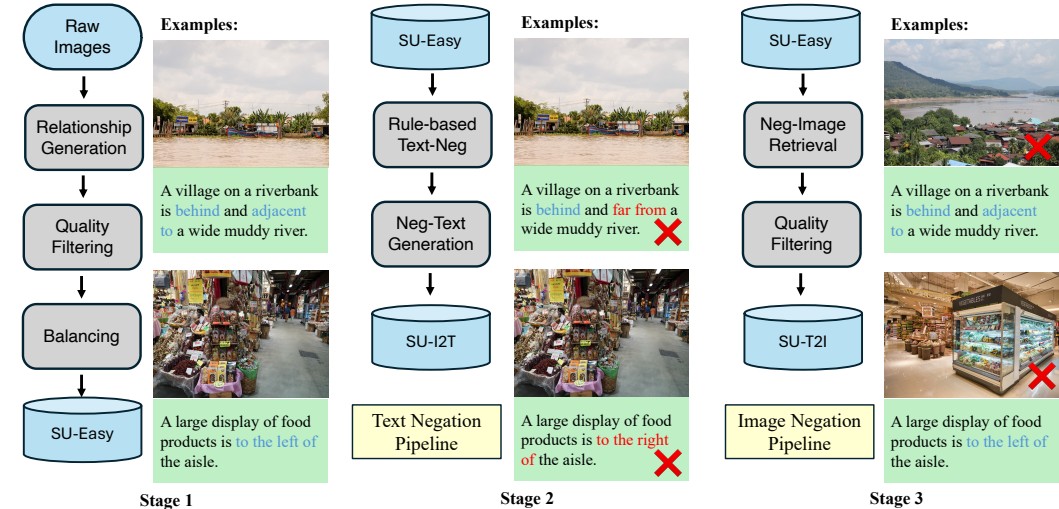

Figure 3: **The Benchmark Construction Pipeline.** SUBench is created with three stages. Stage 1 creates a base dataset (SU-Easy) of positive image-text pairs. Stage 2 generates textual negations (Text Negation Pipeline), followed by a Stage 3 which generates image negations (Image Negation Pipeline) to create challenging negative visual examples. X refers to the negated samples. SU is short for SUBench for simplicity.

generates not only positive image-text pairs but also two distinct types of hard negatives—textual and visual—to facilitate a robust evaluation of retrieval models. The implementation of this multi-stage pipeline is highly engineered. For a complete breakdown of the technical details and prompts used for LLM in different places, please refer to the Appendix.

**Stage 1: Positive Pair Generation (SUBench-Easy)**

To create our high-quality SUBench-Easy dataset, we start by using a guideline-driven LLM (Sec 3.1) to generate captions describing spatial relationships in raw images. By using the LLM again as the autorater, we filter out noisy or ambiguous LLM-generated pairs to ensure precision. Finally, we balance the data by sampling to create a uniform distribution across relationship categories, mitigating model bias towards common terms. This process results in the foundational SUBench-Easy dataset. We name it this way since explicit negative examples are not added yet.

**Stage 2: Text Negation Pipeline (SUBench-I2T)**

To create the SUBench-I2T evaluation set, we generate hard textual negatives from the positive pairs. We programmatically negate spatial relationships in the captions by swapping prepositions with their antonyms (e.g., "left" becomes "right"; "near" becomes "far"). This creates a caption that is semantically close to the original but factually incorrect for the image. These hard negatives are challenging because they retain all the correct objects, forcing a model to rely on a precise understanding of the spatial term to identify the mismatch. After that, we generate a natural language caption based on the spatial labels to serve as the negated caption.

**Stage 3: Image Negation Pipeline (SUBench-T2I)**

To create the SU-T2I evaluation set, we generate hard visual negatives. For each caption from a positive pair, we employ an internal embedding model to retrieve new images that are visually similar to the original but spatially contradicts the caption's description. The obtained negative images are further filtered by using LLM as the rater to filter the images with exactly the same spatial relationship, the final resulting pairs challenge a model's fine-grained visual discrimination, forcing it to differentiate between nearly identical scenes based on subtle spatial changes.

### 3.3 SPATIAL UNDERSTANDING ENHANCEMENT VIA SUPERVISED FINE-TUNING

Our benchmark provides a robust tool for evaluating spatial understanding for VLMs. In addition, we also propose a fine-tuning method designed to explicitly enhance this capability in vision-

language models. Standard contrastive learning objectives, while effective for learning general image-text alignments, often fail to provide a strong enough signal for the model to grasp nuanced spatial relationships. To address this, our method introduces a direct supervision signal by leveraging the standardized spatial labels generated during our benchmark's construction.

Our approach augments a standard vision-language backbone, which consists of a visual encoder and a textual encoder, with a specialized spatial decoder. The overall architecture is depicted in Figure 4. The spatial decoder takes the image and text embeddings as input and generates the class embedding based on them. Note that here the text embeddings are different from those used in contrastive loss. Instead, we remove the spatial-related words from the texts (therefore only objects A and B remain in the text as illustrated in Figure 4) to avoid information leakage.

These embeddings are then passed to the spatial decoder, which is designed to fuse information from both modalities through cross-attention mechanisms and contains only 1 transformer layer. The decoder processes the joint representation and outputs a final hidden state corresponding to a special classification token, `[CLS2]`. This `[CLS2]` token is then fed into a linear classification head to predict the specific spatial relationship label for the given image-text pair (e.g., "above", "left of", "behind"). We have provided the full list of classes in Table A4. This `[CLS2]` token thus serves as a holistic representation of the spatial relationship between the provided objects, with a focus on their relational content.

In addition to the standard contrastive loss $L_{\text{contrastive}}$, the model is supervised by a standard binary cross-entropy loss, which we term it the spatial classification loss ($L_{\text{spatial}}$). This loss is calculated between the predicted probability distribution over all possible spatial labels and the one-hot encoded ground-truth label from our benchmark. We use the standard contrastive loss as standard CLIP (Radford et al., 2021) on the visual `[CLS]` token and pooled text embedding, which are not related to the spatial decoder. By weighting the spatial classification loss with $\lambda$, the final loss is:

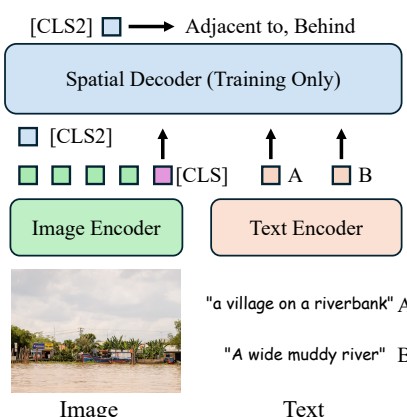

Figure 4: **Overview of the model.** Here we omit the drawing of contrastive learning part and focus on the key components introduced specifically for spatial understanding tasks.

$$L = L_{\text{contrastive}} + \lambda L_{\text{spatial}}.$$

During the fine-tuning stage, this $L_{\text{spatial}}$ provides a powerful and targeted supervisory signal. By optimizing this objective, the gradients are backpropagated through the classification head, the spatial decoder, and ultimately into the visual `[CLS]` token, which is used to do the retrieval. In practical, the spatial decoder is warmed up with N steps for a robust learning process. This joint learning forces the backbone encoders to produce representations that are not only aware of the objects present but are also highly sensitive to their specific spatial configuration. In essence, the model is explicitly trained to ground abstract spatial prepositions from the text within the visual context of the image, thereby improving its foundational spatial understanding for downstream retrieval tasks.

## 4 EXPERIMENTS

In this section, we evaluate the performance of existing VLMs and demonstrate the effectiveness of finetuning with our proposed spatial decoder, which was introduced in the previous section. We provide details of the datasets in Section 4.1 and the experimental setup in Section 4.2.

### 4.1 DATASETS

We utilize a combination of two major datasets in our experiments. The first dataset, WebLI (Wang et al., 2025), is a large-scale dataset for pre-training. The second dataset, Segment Anything 1 Billion (SA-1B) (Kirillov et al., 2023b), serves as the source for our proposed benchmark, SUBench.

**WebLI:** The Web Language Image (WebLI) (Wang et al., 2025) dataset is a web-scale collection of billions of image-text pairs sourced from the public web. We use the same subset of this data created in TIPS (Maninis et al., 2025) to pre-train our CLIP model.

**SA-1B:** SA-1B (Kirillov et al., 2023a) is a dataset designed to train general-purpose object segmentation models. It consists of over 11 million diverse, high-resolution images and 1.1 billion high-quality segmentation masks. Though we do not need the masks, we use SA-1B due to the accessibility of the large-scale, high-resolution, and diverse images.

**SUBench Dataset:** Based on the SA-1B dataset, we constructed two distinct subsets. The first is a split of 1 million images whose corresponding spatial relationships are synthesized by Gemini 2.5 Pro. The second subset (about 10 million) is used to build the image pool for the hard-negative mining in stage 3 of the pipeline. The statistics of SUBench are shown in Table 1. We would like to highlight that the scale of SUBench is much larger than previously used benchmarks. For the training split **SUBench-Training** used for finetuning experiments, we selected $600k$ examples from the 1 million subset excluding the selected $50k$ SUBench examples to avoid leakage. The selection is

| Dataset | Positive | | Negative | |
|---|---|---|---|---|
| | Images | Texts | Images | Texts |
| DOCCI (Onoe et al., 2024) | 14,847 | 14,847 | 0 | 0 |
| MSCOCO (Chen et al., 2015) | 5000 | 5000 | 0 | 0 |
| Flickr30k (Young et al., 2014) | 1000 | 1000 | 0 | 0 |
| SUBench-Easy (Ours) | 50,000 | 50,000 | 0 | 0 |
| SUBench-T2I (Ours) | 50,000 | 50,000 | 110,276 | 0 |
| SUBench-I2T (Ours) | 50,000 | 50,000 | 0 | 79,000 |

Table 1: Statistics for commonly used image-text retrieval datasets and SUBench.

**Benchmark Quality:** We assess SUBench data quality via a human validation study (Table 2) on 100 randomly sampled instances, where two independent annotators evaluated whether each caption is aligned with the corresponding image in terms of visual and spatial content. We observe only 3% visual and 8% spatial discrepancies, with limited overlap on the latter (2 of 8 cases flagged by both reviewers), suggesting that many potential "errors" stem from scene ambiguity or subjective interpretation rather than clear annotation mistakes. Overall, the low error rate indicates that SUBench exhibits high fidelity and is suitable for reliable evaluation.

| Reviewer | Visual Feature Error | Spatial Feature Error |
|---|---|---|
| R1 | 1 | 6 |
| R2 | 3 | 4 |
| Total | 3 | 8 |
| Common | 1 | 2 |

Table 2: Human verification of errors on 100 sampled instances from SUBench.

## 4.2 EXPERIMENTAL SETUPS

Our experimental setup involves two key stages: pre-training and fine-tuning. We use a CLIP model architecture as our base model, which consists of a Vision Transformer (ViT) (Dosovitskiy et al., 2021) image encoder and a Transformer-based text encoder. Details of the hyperparameters can be found in Appendix E.

**Pre-training:** In the first stage, we pre-train CLIP model using the standard contrastive learning objective on the WebLI dataset. This process aligns the image and text embedding spaces and establishes a strong baseline for general vision-language understanding. Note that for simplicity, we do not use the advanced techniques for CLIP such as SigLIP2 (Tschannen et al., 2025).

**Fine-tuning:** We finetune the pre-trained CLIP baseline model on a mix of the WebLI dataset and SUBench-Training dataset with an equal ratio. This co-training strategy ensures that the model

continues to learn from a broad distribution of web-scale data while simultaneously specializing in spatial reasoning. Furthermore, during this stage, we introduce an additional spatial classification loss on SUBench-Training data, which specifically penalizes incorrect spatial predictions and makes the model more sensitive to relational terms.

## 4.3 BENCHMARK RESULTS

| Model | T2I Retrieval | | | | | I2T Retrieval | | | | | CLS |
|---|---|---|---|---|---|---|---|---|---|---|---|
| | MSCOCO | Flickr30k | DOCCI | SU-Easy | SU-T2I | MSCOCO | Flickr30k | DOCCI | SU-Easy | SU-I2T | ImageNet |
| CLIP-B/16-224 | 33.1 | 62.1 | - | 32.3 | 17.1 | 52.4 | 81.9 | - | 35.5 | 15.2 | 68.3 |
| SigLIP-B/16-224 | 47.2 | 77.9 | - | 49.3 | 27.0 | 64.5 | 89.6 | - | 49.9 | 19.7 | 76.2 |
| SigLIP2-B/16-224 | 52.1 | 80.7 | - | 46.3 | 24.4 | 68.9 | 93.0 | - | 45.2 | 18.5 | 78.2 |
| TIPS-B/14-448 | 54.7 | 79.4 | 50.7 | 41.2 | 23.8 | 69.1 | 91.3 | 50.3 | 47.1 | 20.9 | 76.3 |
| TIPS-g/14-448 | 59.2 | 83.8 | 59.4 | 45.8 | 27.3 | 74.0 | 93.8 | 57.0 | 45.5 | 23.0 | 79.7 |

Table 3: **Benchmark Results.** We benchmark the advanced VLMs on the retrieval tasks. We report Recall@1 for image-to-text (I2T) and text-to-image (I2T) retrieval. We also report zero-shot classification (CLS) result on the ImageNet dataset.

We benchmark several state-of-the-art models against our proposed benchmark, with full results in Table 3. Our findings highlight a critical vulnerability: models that achieve impressive results on traditional image-text retrieval benchmarks show a low performance on SUBench. More specifically, the substantial performance gap between the SUBench-Easy subset and the hard negative subsets (SUBench-T2I and SUBench-I2T) reveals the core of the issue. Current VLMs heavily rely on matching general semantic content; their retrieval capabilities collapse when presented with our hard negatives, where semantics are nearly identical to the positive pair and only the spatial relationship is incorrect. Furthermore, we observe that a generally "stronger" model does not guarantee better performance on these challenging spatial tasks, indicating that the nuanced spatial awareness tested by SUBench is a specialized skill not effectively captured during standard pre-training. Therefore, there is need to improve the spatial understanding capability of VLMs with specific design.

## 4.4 RESULTS AND DISCUSSION

We show several visualizations in Figure 5, demonstrating that our model, trained with the spatial classification loss, has a much better understanding of complex spatial relationships than the strong TIPS-g/14-448 baseline. For instance, we observe that the baseline model struggles with perspective-related retrieval; when prompted for "umbrellas ... above and in front of a large body of water," it incorrectly retrieves an image of large trees, confusing the semantic and spatial context. Our model, however, correctly retrieves an image matching the prompt. Furthermore, existing VLMs tend to ignore topological relationships like contains. As seen in the bottom-right example, the baseline retrieves an empty hall despite the prompt specifying it "contains a group of people," a detail our model accurately captures. These results underscore our model's ability to ground language in visual scenes with greater fidelity.

## 4.5 ABLATION STUDY

To understand the individual impact of our key contributions, we conduct an ablation study on the fine-tuning stage. The results are summarized in Table 4, where we report Recall@1 on SUBench, as well as performance on general image-text retrieval benchmarks like MSCOCO (Chen et al., 2015) and Flickr30k (Young et al., 2014) to evaluate potential catastrophic forgetting.

**Ablation on training strategy.** Table 4 analyzes how different training strategies affect both standard retrieval benchmarks and SUBench. All models in the bottom block share the same CLIP-B backbone; we also report off-the-shelf CLIP from OpenAI (Radford et al., 2021) and a much larger specialist model, TIPS-g/14-448 (Maninis et al., 2025), as external references. Off-the-shelf CLIP performs reasonably on MSCOCO and Flickr30k, but is very weak on SUBench (17.1/15.2 Recall@1 on SU-T2I/SU-I2T). TIPS-g substantially improves spatial retrieval but benefits from a significantly larger architecture and training recipe. Starting from our Baseline model, which is trained on generic WebLI-style data and already improves over CLIP-B on MSCOCO/Flickr30k/DOCCI,

| **TIPS-g/14-448** | **Ours** | **TIPS-g/14-448** | **Ours** |

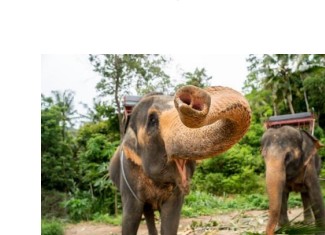 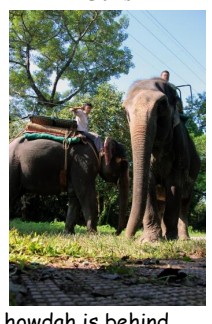 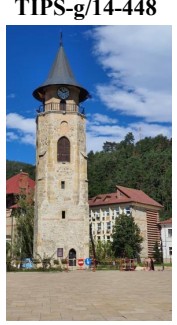 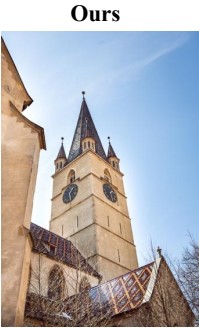

An elephant with a green and red howdah is behind and to the left of an elephant with its trunk pointing towards the ground.

A tall stone clock tower with a pointed steeple is above and behind a roof with colorful, patterned tiles.

| **TIPS-g/14-448** | **Ours** | **TIPS-g/14-448** | **Ours** |

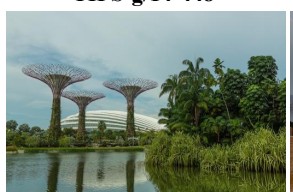 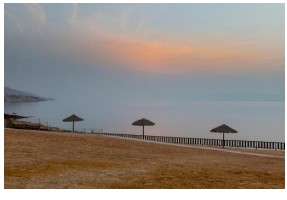 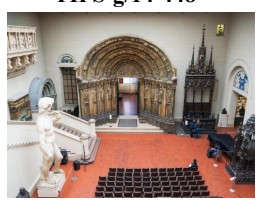 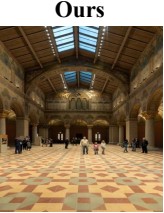

A group of three thatched umbrellas is positioned above and in front of a large body of water.

A large, ornate hall contains a group of people.

Figure 5: **Qualitative Comparison of Text-to-Image Retrieval Results**. For each text prompt, we show the top image retrieved by the strong TIPS-g/14-448 baseline and our model with $L_{\text{spatial}}$. Our model demonstrates a superior understanding of complex spatial relationships and compositional details, retrieving images that more accurately match the given prompts.

| Model | T2I Retrieval | | | | | I2T Retrieval | | | | | CLS |
|---|---|---|---|---|---|---|---|---|---|---|---|
| | MSCOCO | Flickr30k | DOCCI | SU-Easy | SU-T2I | MSCOCO | Flickr30k | DOCCI | SU-Easy | SU-I2T | ImageNet |
| CLIP-B/16-224 | 33.1 | 62.1 | - | 32.3 | 17.1 | 52.4 | 81.9 | - | 35.5 | 15.2 | 68.3 |
| TIPS-g/14-448 | 59.2 | 83.8 | 59.4 | 45.8 | 27.3 | 74.0 | 93.8 | 57.0 | 45.5 | 23.0 | 79.7 |
| Baseline | 37.9 | 65.4 | 32.8 | 29.3 | 14.0 | 53.1 | 81.4 | 31.3 | 29.3 | 12.6 | 71.7 |
| + SUBench-Training | 23.8 | 47.8 | 27.7 | 47.4 | 30.2 | 38.0 | 69.2 | 29.7 | 47.4 | 24.8 | 50.6 |
| + Co-training | 38.6 | 66.8 | 40.7 | 52.9 | 35.0 | 56.8 | 83.9 | 41.8 | 52.0 | 25.7 | 71.7 |
| + $L_{\text{spatial}}$ (Ours) | 39.0 | 67.8 | 42.1 | **56.3** | **37.1** | 57.3 | 85.2 | 43.2 | **55.9** | **28.2** | 71.6 |

Table 4: **Ablation Studies on the Model Training.** We report Recall@1 for retrieval benchmarks. We also report zero-shot classification (CLS) result on the ImageNet dataset. The highest evaluation values in SUBench are highlighted in **bold** texts.

naively fine-tuning only on SUBench (+ *SUBench-Training*) strongly boosts spatial retrieval (SU-T2I $14.0 \rightarrow 30.2$, SU-I2T $12.6 \rightarrow 24.8$), but severely degrades general retrieval and ImageNet zero-shot classification. This confirms that directly overfitting to our benchmark distribution can hurt overall visual-language capabilities. Co-training SUBench with the original WebLI data (+ *Co-training*) mitigates this issue: we recover and even improve MSCOCO/Flickr30k/DOCCI performance while further increasing SUBench scores (SU-T2I 35.0, SU-I2T 25.7), and fully restore ImageNet accuracy to the baseline level. Finally, adding our spatial loss $L_{\text{spatial}}$ (+ $L_{\text{spatial}}$ *(Ours)*) yields the best overall performance. Compared to the co-training model, we observe consistent gains on all SUBench metrics (SU-T2I $35.0 \rightarrow 37.1$, SU-I2T $25.7 \rightarrow 28.2$), without sacrificing ImageNet accuracy and standard retrieval benchmarks. This shows that explicit spatial supervision not only enhances spatial retrieval, but can be incorporated without harming (and in some cases even mildly improving) general-purpose representation quality.

**Effect of spatial re-captioning at scale.** Table 5 studies the effect of applying our spatial captioning pipeline to a 4M-image subset of WebLI. Starting from our best model (*Ours (SU-Training + Raw)*), adding an additional data source with SUBench-style spatial captions (+ *WebLI 4M Recap*)

| Model | T2I Retrieval | | | | | I2T Retrieval | | | | | CLS |
|---|---|---|---|---|---|---|---|---|---|---|---|
| | MSCOCO | Flickr30k | DOCCI | SU-Easy | SU-T2I | MSCOCO | Flickr30k | DOCCI | SU-Easy | SU-I2T | ImageNet |
| Baseline | 37.9 | 65.4 | 32.8 | 29.3 | 14.0 | 53.1 | 81.4 | 31.3 | 29.3 | 12.6 | 71.7 |
| Ours (SU-Training + Raw) | 39.0 | 67.8 | 42.1 | 56.3 | 37.1 | 57.3 | 85.2 | 43.2 | 55.9 | 28.2 | 71.6 |
| + WebLI 4M Recap | 38.5 | 65.8 | 42.5 | 59.9 | 41.6 | 57.3 | 83.3 | 43.6 | 59.4 | 35.6 | 71.9 |
| Raw + WebLI 4M Recap | 39.8 | 68.0 | 44.4 | 45.9 | 30.3 | 56.3 | 83.6 | 44.2 | 46.0 | 24.0 | 73.0 |

Table 5: **Ablation Studies on the Data Scaling.** We report Recall@1 for retrieval benchmarks. We also report zero-shot classification (CLS) result on the ImageNet dataset. Raw refers to raw WebLI dataset without recaption used in the Baseline and Co-Training settings.

leads to further improvements on SUBench, with SU-T2I increasing from 37.1 to 41.6 and SU-I2T from 28.2 to 35.6. At the same time, performance on MSCOCO, Flickr30k, DOCCI, and ImageNet remains comparable or slightly better, indicating that enriching large-scale pre-training data with spatially explicit captions does not trade off general retrieval performance. To isolate the effect of spatial re-captioning without any direct SUBench-Training supervision, we also train a model using only Raw WebLI and 4M re-captioned WebLI (*Raw + WebLI 4M Recap*). Even in this setting, the model achieves substantially higher SU-T2I performance (30.3 Recall@1) than the off-the-shelf CLIP-B and our original Baseline trained on generic captions, while preserving or slightly improving standard retrieval and classification accuracy. Together, these results suggest that our spatial captioning pipeline is broadly useful beyond SUBench: injecting spatial descriptions into large-scale image–text corpora systematically enhances spatial retrieval, while maintaining strong general vision–language performance.

## 5 CONCLUSION

In this work, we introduce SUBench, a benchmark contains $50k$ image-text pairs and hard negative texts and images. SUBench focuses on the spatial understanding in the cross-modal retrieval task. Our benchmark was curated using a LLM-based framework that aligns objective spatial relationships with subjective human descriptions, all guided by a principled taxonomy that reduces ambiguity. Our experiments reveal that even state-of-the-art VLMs struggle on SUBench, highlighting a critical blind spot in their spatial understanding capabilities. We also demonstrate that by finetuning a model on a training set curated with the same methodology, we observed substantial performance gains on SUBench and improved generalization on existing retrieval benchmarks.

We acknowledge several limitations that open avenues for future research:

**Generalization to New Sources.** While this work curated SUBench from the SA-1B dataset, the proposed LLM-based pipeline is fundamentally dataset-agnostic. This framework can be applied to arbitrary image collections to generate new spatial-aware training and evaluation data. Crucially, the integrated quality filtering process is designed to ensure robustness, automatically removing noisy, ambiguous, and trivially true cases to maintain the high quality of the generated data regardless of the image source.

**Static Worldview.** SUBench is based on static images and thus cannot evaluate a model's understanding of dynamic or temporal spatial relationships (e.g., "the car is moving away from the house"). Future work could extend this framework to video data to create benchmarks for more complex, real-world scenarios.

**Taxononomy Scope.** While our taxonomy of spatial relations is principled, it is not exhaustive. It could be expanded to include more nuanced concepts such as occlusion, complex containment, and orientation-dependent relations that are vital for robotics and embodied AI.

**LLM-Based Data Curation vs Human annotation.** Our scalable pipeline relies on an LLM for data generation and verification. While effective, this process may inherit subtle biases or generate occasional errors. Future research could explore multi-LLM verification systems or human-in-the-loop pipelines to further refine data quality.

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

# A    DATA PIPELINE

In this section, we provide more details on the pipeline discussed in Section 3.2.

## A.1    STAGE 1 GENERATION

```
relationship:
  reason: ...
  label: ... # e.g., VDR-0/TR-0
  caption: ...
  confidence_score: ... # 0-5
object_a_description: ...
object_b_description: ...
```

Figure A1: The structured output format from our data generation pipeline.

We started by feeding one million randomly sampled images from the SA-1B dataset (Kirillov et al., 2023b) to Gemini 2.5 Pro (Deepmind et al., 2025) using a specific generation prompt (Prompt 1.1, see B.1), which was designed to create initial image-text pairs in a structured format that included fields such as relationship, reason, label, caption, confidence score (on a scale of 0-5), and descriptions for two objects. We used structured output format as shown in Figure A1.

| Stage | Count |
|---|---|
| Input Images | 1M |
| Filtered Samples (Removed) | 127k |
| Data Format Errors (Removed) | 3.5k |
| **Remaining Samples** | **869k** |

Table A1: Statistics of the initial data filtering process.

However, we observed that some of the generated descriptions contained errors, such as mentioning an object that was difficult to find due to the presence of other similar objects. To address this, we implemented a data filtering pipeline (Prompt 1.2, see B.2) to automatically identify and remove these low-quality or ambiguous pairs. According to our filtering statistics (Table A1), this stage removed 127k samples, with an additional 3.5k removed due to data format errors, leaving 869k samples. After this cleaning process, we balanced the dataset based on the relationship categories assigned by Gemini to ensure a fair distribution. This entire procedure culminated in our final set of 50k reliable image-text pairs, which forms the easy version of our benchmark (SUBench-Easy).

## A.2    STAGE 2: HARD TEXTUAL NEGATIVE GENERATION

| Category | Label | Name | Negated-Label | Negated-Name |
|---|---|---|---|---|
| Vertical Directional | VDR-0 | above | VDR-1 | below |
| Anteroposterior Directional | ADR-0 | in front of | ADR-1 | behind |
| Lateral Directional | LDR-0 | left | LDR-1 | right |
| Proximity Directional | PDR-0 | near | PDR-1 | far |
| Topological | TR-0 | within | TRDR-2 | apart from |
| Topological | TR-1 | contains | TRDR-2 | apart from |
| Distributional | DR-0 | amid | TRDR-2 | apart from |
| Distributional | DR-1 | around | TRDR-2 | apart from |

Table A2: Mapping between labels (names) and their negative labels (names) for different categories.

The second stage of our pipeline is designed to create the SUBench-I2T (Image-to-Text) evaluation set by generating hard textual negatives. This process takes the validated positive pairs from Stage 1 and systematically manipulates the textual caption to create a factual mismatch with the corresponding image, thereby testing a model's fine-grained comprehension.

The core of this stage is a rule-based negation engine. We apply a set of deterministic rules to build the negated text. This targeted substitution alters the core spatial relationship while leaving all other semantic elements. A summary of these negation rules is provided in Table A2. After we got the negated label, we use Gemini 2.5 Pro to generate (Prompt 2.1, see B.3) a natural language to serve as the negated text.

### A.3 STAGE 3: HARD IMAGE NEGATIVE GENERATION

| Metric / Filtering Step | Count |
|---|---|
| Initial Pool of Positive Captions | 50k |
| **Candidate Image Retrieval and Filtering** | |
| Total Candidate Images Retrieved (with A and B) | 500k |
| *Rejected by Prompt 3.1: Object Detectability* | |
| Images Rejected (Cannot Find Objects A or B) | 293k |
| *Rejected by Prompt 3.1: Relationship Mismatch* | |
| Images Rejected (Relationship Is Same) | 97k |
| **Final Hard Image Negatives Generated** | 110k |
| **Dataset Coverage** | |
| Positive Captions with $\geq 1$ Hard Negative Image | 28.3k |
| % Coverage of Initial Positive Captions | 56.6 % |

Table A3: Summary of hard image negative generation (Stage 3) and filtering process.

The third stage of our pipeline focuses on generating hard image negatives for the SUBench-T2I (Text-to-Image) evaluation set. This process requires a different approach from the textual negation in Stage 2, as it involves finding an image that intentionally violates the spatial relationship described in the original caption, while still containing the same core objects. To achieve this, we first retrieve a large pool of candidate images from a large-scale image dataset (SA-1B, excluding the original positive 1M image set) that contain both object A and object B as described in the original text. After we retrieved the images, we apply a filtering prompt (Prompt 3.1, see B.4) to Gemini 2.5 Pro. This prompt is designed to verify two critical conditions: first, that both objects are findable within the image, and second, that the spatial relationship between them, as described in the original text, is factually false in the retrieved image. Only images that meet these criteria are selected as hard image negatives, creating a challenging benchmark for models to distinguish between subtle visual differences. We calculate the statistics of the data filtering in Table A3.

## B PROMPT DESIGN

In this section, we describe the details about the prompt used in the three stages. Note that the original prompts are too verbose and we only put the core part in the paper.

### B.1 PROMPT 1.1

**Guidelines.** The prompt begins with a set of overarching guidelines that govern the entire process. These are not optional suggestions but are non-negotiable constraints designed to ensure the quality and consistency of the output. Key principles include: *Precision, Adherence, and Consistency*: The output must strictly follow all rules and definitions. *Exclusive Vocabulary*: The AI is mandated to use only the predefined relationship labels (VDR-0, ADR-0, etc.), preventing any creative or ad-hoc descriptions in the final label. *Location-Independent Object Descriptions*: Objects must be identified without using spatial location (e.g., "a man on the left" is forbidden), which forces the AI to focus

on intrinsic object properties. *The Salience Principle*: This is a critical rule that instructs the AI to avoid generic, trivially true relationships (e.g., "a building is on the ground") and instead identify relationships that are specific and meaningful to the image's content.

| Category | Relationship | Description | Synonyms |
|---|---|---|---|
| Vertical Directional | above (VDR-0) | A is at a higher vertical plane than B. | on top of, over, elevated from, superior to, higher than. |
| | below (VDR-1) | A is at a lower vertical plane than B. | underneath, beneath, under, inferior to, situated lower than |
| Anteroposterior Directional | in front of (ADR-0) | A is closer to the viewer than B. | before, ahead of, preceding, facing |
| | behind (ADR-1) | A is further from the viewer than B. | at the rear of, after, following, to the back of |
| Lateral Directional | left (LDR-0) | A is to the left of B from the viewer's perspective. | to the left of, on the left side of |
| | right (LDR-1) | A is to the right of B from the viewer's perspective. | to the right of, on the right side of |
| Proximity Directional | near (PDR-0) | A is in close spatial proximity to B. | close to, adjacent to, beside, next to, nearby, proximate to. |
| | far (PDR-1) | A is at a significant distance from B. | distant from, remote from, a long way from |
| Topological | within (TR-0) | A is enclosed by the boundaries of B. | inside, in, encompassed by, enclosed by, contained in |
| | contains (TR-1) | A encloses B. (Inverse of within) | holds, encloses, encompasses, includes, has inside |
| Distributional | amid (DR-0) | A is in the middle of a collection of B. | amongst, in the midst of, surrounded by, in the middle of |
| | around (DR-1) | A is circling or surrounding B. | circling, surrounding, encircling, encompassing |

Table A4: **Allowed Spatial Relationships and Definitions.**

**Allowed Spatial Relationships and Definitions.** The prompt defines a precise and hierarchical taxonomy of spatial relationships. This is the core of the AI's "knowledge" for this task. The relationships are categorized and assigned unique labels, ensuring a standardized output. We also provide synonyms to let the AI to use them in the caption to keep the subtitles varied and more natural-sounding, which ultimately enhances the overall quality of the generated captions.

**Examples** The prompt provides concrete examples to illustrate the nuances of the defined relationships, particularly for ambiguous or easily confused concepts. Examples for amid vs. around (DR-0 vs. DR-1) and near vs. far (PDR-0 vs. PDR-1) help the AI distinguish between these relationships. A key takeaway from the examples is the emphasis on prioritizing topological (TR) and distributional (DR) relationships when they apply, as they often convey more specific information than simple directional ones.

**Compound Labels.** A compound label is a combination of up to two single relationship labels from different categories. This feature allows the AI to capture more complex spatial relationships that a single label cannot adequately describe. For instance, an object can be both above and far from another object. The prompt specifies a strict ordering rule for these combinations: VDR followed by ADR, then LDR, PDR, TR, and DR. This means VDR-0/LDR-0 (e.g., "above and left") is a valid combination, but LDR-0/VDR-0 is not. This rule ensures the output is both expressive and consistently formatted.

**Frames of Reference.** This is arguably the most critical component for ensuring the consistency of the output. The prompt strictly defines and prioritizes the use of a Relative Frame (Egocentric/Viewer-Centric) for all directional relationships (VDR-x, ADR-x, LDR-x). This means the AI must always interpret "left," "right," "above," etc., from the perspective of the image's viewer. The prompt explicitly forbids the use of other frames, such as: *Intrinsic Frame (Object-Centric)*: Relationships based on an object's internal orientation (e.g., "the car's engine is in front of its trunk"). *Absolute Frame (World-Centric/Cardinal)*: Relationships based on fixed coordinates (e.g., "the building is south of the park"). The prompt's design requires the AI to recognize when an alternative frame might be considered by a human and explicitly state why the Relative Frame was chosen, thereby preventing common errors in spatial reasoning.

**Scoring System.** The prompt includes a confidence scoring system from 0 to 5. This provides a measure of certainty for each generated relationship. A score of 5 is reserved for relationships where both objects are prominent and the relationship is unambiguous. This scoring mechanism is a form of self-evaluation, allowing the AI to indicate the reliability of its output, which can be valuable for downstream data filtering or analysis.

**Output Format.** The final output is a YAML object, which provides a structured and easily parsable format for the data.

## B.2 PROMPT 1.2

**Guideline.** The verification task is to rigorously verify the accuracy, adherence to rules, and overall quality of spatial relationship descriptions produced by another AI model. The AI model has access to the original image, the full set of instructions given to the generating model, and the output of the generating model.

**Object Description Validity.** Based on the prompt, the AI system must verify that the descriptions of the two objects are accurate, present in the image, and specific enough to uniquely identify each object. Descriptions should avoid using spatial location (e.g., "the person on the left") and use articles correctly.

**Spatial Relationship Accuracy & Adherence.** The prompt requires the AI system to confirm that the described spatial relationship is factually correct and that its associated label (e.g., VDR-0 for "above") aligns with the provided definitions and vocabulary. It is critical that directional relationships are interpreted exclusively from a viewer-centric perspective.

**Relationship Unambiguity, Strength, & Completeness.** The AI system must check that the relationship is unambiguous, a clear and strong representation of the spatial connection (e.g., "above" means distinctly above, not just "slightly higher"), and fully describes all relevant positional aspects (e.g., if a compound relationship like "above and left" is more accurate than "above").

**YAML Output.** The final output is required to be a structured YAML object, ensuring the data is easily parsable and consistently formatted. The output also includes a confidence score (from 0 to 5) to indicate the reliability of the model's assessment.

## B.3 PROMPT 2.1

This section details a systematic prompt-based methodology for generating negated spatial relationship data. The core function is to take a given spatial relationship between two objects and produce its logical negation(s). The process is governed by a set of rules that depend on the semantic category of the relationship.

| Hyperparameter | Pre-training | Fine-tuning |
|---|---|---|
| Model Size | ViT-B/14 | |
| Resolution | $224 \times 224$ | |
| Patch Size | $14 \times 14$ | |
| Hidden Dim (D) | 768 | |
| Num Heads | 12 | |
| Num Layers | 12 | |
| Optimizer | AdamW | |
| Optimizer $\beta$s | (0.9, 0.999) | |
| Batch Size | 16384 | |
| Learning Rate | $5 \times 10^{-4}$ | |
| LR Schedule | Linear Decay | |
| Weight Decay | 0.01 | |
| Loss Function | Contrastive Loss | |
| Warmup Steps | 10k | 5k |
| Iterations | 250k | 10k |

Table A5: Hyperparameter settings for the pre-training and fine-tuning stages. The configuration follows standard practices for training Vision Transformer models.

**Swap-Negatable Relationships.** For relationships that exist on a spectrum with a clear opposite, negation is performed through a direct swap. This category includes Vertical (above/below), Antero-posterior (in front of/behind), Lateral (left/right), and Proximity (near/far) directional relationships. The process involves two simultaneous changes: The relationship's label is swapped with its inverse counterpart (e.g., VDR-0 becomes VDR-1). The corresponding term in the caption is replaced with its antonym (e.g., "above" is changed to "below"). This approach maintains the specific dimensional context of the original relationship while inverting its direction.

**Non-Swap-Negatable Relationships** This category covers Topological (within, contains) and Distributional (amid, around) relationships, which lack direct, one-to-one antonyms within the defined taxonomy. Negating these relationships involves a different mechanism. Instead of swapping to an opposite, the negation asserts a general state of separation.The original label (e.g., TR-0 for "within") is replaced with a dedicated negation label, TRDR-2. The caption is rewritten to use the phrase "apart from," which serves as a universal negative for these specific spatial conditions. This design choice prevents semantic ambiguity, as the negation of "within" is not necessarily "contains," but more broadly, "not within".

**Handling Compound Relationships** The prompt is designed to handle complex spatial descriptions that combine multiple relationships (e.g., "above and to the left of"). When presented with a compound label such as VDR-0/LDR-0, the system does not negate the entire phrase at once. Instead, it decomposes the relationship and negates each component individually, generating a separate output for each negation. For the input VDR-0/LDR-0 ("above and to the left of"), two distinct negated outputs are generated: (1) Negating the vertical component: VDR-1/LDR-0 ("below and to the left of"); (2) Negating the lateral component: VDR-0/LDR-1 ("above and to the right of").

## B.4 PROMPT 3.1

**Core Task.** The prompt is designed to determine if the spatial relationship between two specific objects changes across a pair of images.

**Two-Step Verification.** The prompt requires the AI system to determine: (1) *Object Presence Check* and (2) *Relationship Change Check*. The *Object Presence Check*: First, it must confirm if both specified objects are present in the second image. The analysis stops if either is missing. *Relationship Change Check*: Second, if both objects are present, it must determine if their spatial arrangement in the second image is different from the first.

**Strict Frame of Reference.** The most critical rule is the mandatory use of the Relative (Viewer-Centric) Frame for all directional relationships (above, left, in front of, etc.). This ensures consistency by an external observer's point of view.

| | VDR | ADR | LDR | PDR | TR | DR | VDR/ADR | VDR/LDR | VDR/PDR | ADR/LDR | ADR/PDR | LDR/PDR | Sum |
|---|---|---|---|---|---|---|---|---|---|---|---|---|---|
| Count | 3000 | 3000 | 3000 | 0 | 6000 | 6000 | 4500 | 4000 | 4500 | 6000 | 6000 | 4000 | 50000 |

Table A6: Count of samples for each category.

| | T2I Recall@1 (%) | | | | | | | I2T Recall@1 (%) | | | | | | |
|---|---|---|---|---|---|---|---|---|---|---|---|---|---|---|
| Model | ADR | LDR | VDR | PDR | DR | TR | Total | ADR | LDR | VDR | PDR | DR | TR | Total |
| Baseline | 14.0 | 9.4 | 13.9 | 14.0 | 14.4 | 20.3 | 14.0 | 11.8 | 8.0 | 10.9 | 11.5 | 16.7 | 18.8 | 12.6 |
| SUBench-Training | 33.5 | 26.2 | 29.8 | 31.5 | 28.4 | 32.7 | 30.2 | 25.9 | 17.2 | 20.7 | 24.1 | 32.0 | 31.6 | 24.8 |
| + Co-training | 38.4 | 30.3 | 34.7 | 36.3 | 32.4 | 38.5 | 35.0 | 27.7 | 18.5 | 22.9 | 25.1 | 30.5 | 31.2 | 25.7 |
| + Spatial | 40.8 | 32.7 | 36.6 | 38.2 | 35.0 | 40.7 | 37.1 | 30.5 | 21.2 | 24.8 | 27.9 | 31.7 | 33.9 | 28.2 |

Table A7: **Fine-grained Recall@1 on SUBench by relation type.** ADR: Anteroposterior Directional; LDR: Lateral Directional; VDR: Vertical Directional; PDR: Proximity Directional; DR: Distributional Directional; TR: Topological Directional.

**Input.** Different from previous prompts, we give the AI system both the original image and the retrieved image. This is to ensure the AI system can unambiguously understand the spatial relationship.

## C    FINE-GRAINED ANALYSIS BY RELATION TYPE

In the main paper we report overall retrieval performance on SUBench. Here, we provide a more fine-grained analysis broken down by relation type. Specifically, we report Recall@1 for six relation categories (ADR, LDR, VDR, PDR, DR, TR) for both text-to-image (T2I) and image-to-text (I2T) retrieval, under four training configurations: the baseline (**Baseline**), training on SUBench only (**SUBench-Training**), joint training with the original data (**+ Co-training**), and our full model with the proposed spatial loss (**+ Spatial**). As shown in Table A7, our methods consistently improve performance over the Baseline across all relation categories. Starting from the Baseline, SUBench-Training already yields large gains. Further co-training and the spatial-loss model (+Spatial) continue to improve all relation types, pushing T2I overall to 37.1 and I2T overall to 28.2. This indicates that supervised fine-tuning on SUBench, coupled with the proposed spatial loss, yields consistent gains among all relation categories rather than optimizing for a narrow subset. At the same time, the absolute numbers reveal a clear performance ceiling under our current architecture and data. Even for our strongest model (+Spatial), lateral directional relationships (LDR, e.g., left vs. right) remain noticeably weaker than other relations. For example, T2I LDR improves from 9.4 (Baseline) to 32.7 (+Spatial), but still lags behind ADR ($14.0 \rightarrow 40.8$) and TR ($20.3 \rightarrow 40.7$). A similar trend holds for I2T LDR ($8.0 \rightarrow 21.2$) and for vertical directional relationships (VDR). These patterns suggest that fine-grained lateral and vertical reasoning is intrinsically more challenging for current retrieval architectures.

## D    DISCUSSION WITH EXISTING WORKS

**T2I generation models focusing on spatial relationships.** Recent work on spatially faithful text-to-image (T2I) generation includes SPRIGHT (Chatterjee et al., 2024), a large-scale re-captioned dataset designed to improve the spatial consistency of T2I models. SPRIGHT reports gains on generative benchmarks like T2I-CompBench (Huang et al., 2025). These methods primarily aim to make generators follow spatial prompts more faithfully. By contrast, SUBench is built entirely on real images and is used to probe the spatial reasoning of retrieval models under realistic appearance statistics (clutter, occlusion, groups), rather than to evaluate image synthesis quality. In Stage 3 of our pipeline, we retrieve candidate real images sharing the same objects as the positive pair and then apply an LLM verifier. This step discards images where objects are missing or where the spatial relation matches the caption, keeping only truly contradictory hard negatives. More than half of the 500k candidates are rejected in this process, yielding 110k vetted negatives covering 56.6% of positives. We also conducted preliminary experiments with a recent commercial T2I system, Nano Banana, and observed that fine-grained spatial instructions are still non-trivial to enforce reliably

(Figure A10 and A11). This makes it difficult to guarantee that positives and negatives differ *only* in the spatial relation—the key isolation required for a rigorous retrieval benchmark. Consequently, we rely on real image retrieval plus LLM verification in SUBench, viewing T2I-based hard negative synthesis (as explored in TripletCLIP (Patel et al., 2024)) as a complementary direction for future extensions.

**Schemes to improve CLIP with synthetic negatives.** TripletCLIP (Patel et al., 2024) proposes a contrastive pre-training strategy that augments CLIP with synthetic vision–language triplets. In this approach, hard negative captions are produced via in-context prompting, and corresponding negative images are synthesized with modern T2I models; these triplets are then used to train CLIP with a triplet-style objective. While this approach is explicitly designed to improve the CLIP model itsel, our primary contribution is not a new pre-training recipe. Instead, we introduce a benchmark and data pipeline tailored to spatial-related retrieval: SUBench defines spatially controlled positives and hard negatives in a purely real-image regime. This exposes a specific blind spot of existing CLIP-like models: their difficulty in discriminating pairs that differ only in spatial relationships.

**VQA benchmarks for spatial reasoning.** SpatialVLM (Chen et al., 2024a) and related works (Liu et al., 2023; Kamath et al., 2023; Jia et al., 2025; Cheng et al., 2024) target the spatial VQA task by generating massive 3D spatial reasoning datasets to train VLMs capable of answering qualitative and quantitative spatial questions. While our work shares an interest in spatial understanding, SUBench addresses a fundamentally different problem: text–image retrieval rather than VQA. In VQA, the model receives both the image and the question simultaneously, allowing visual context to help resolve ambiguities. In T2I/I2T retrieval, however, a single text (or image) embedding must be sufficiently self-contained and discriminative to select the correct item from a large database. This makes benchmark construction significantly harder, as captions must remain natural yet precise enough to distinguish "hard negatives" that are semantically similar but spatially incorrect. SUBench is specifically designed for this retrieval setting. By providing real-image hard negatives that share objects but contradict the spatial relation in the caption, we reveal that current CLIP-style models—despite strong performance on standard retrieval benchmarks—collapse when only the spatial relationship differs. Our main contribution is the data pipeline and benchmark (SUBench) that isolates this failure mode. The spatial decoder in our method serves as a validation tool to demonstrate that targeted data and losses can close part of this gap, rather than as a standalone architectural contribution.

# E    EXPERIMENT SETTINGS

Our experimental setup is detailed in Table A5. We follow the standard contrastive learning framework for CLIP and, for simplicity, do not employ advanced loss functions or training strategies.

# F    BENCHMARK STATISTICS

Table A6 presents the sample counts for each category. Note that Proximity Directional Relationships (PDR) always appear as part of a compound label.

# G    MORE VISUALIZATION RESULTS

**SUBench Samples.** Figure A2 showcases several examples from the SUBench dataset.

**Filtered Samples.** We provide examples of samples filtered out by Prompt 1.2 (see Section B.2). The samples in Figures A3, A4, and A5 were removed due to incorrect spatial relationships. The sample in Figure A6 was excluded because it described a generic truth.

**Negated Images.** Figures A7, A8, and A9 display samples that were negated and subsequently verified by Prompt 3.1 (see Section B.4).

# H    LLM USAGE

We utilized Gemini 2.5 Pro for two purposes in this work. First, it was used to generate the data for our benchmark. Second, it served as a writing aid to improve the language of this paper by

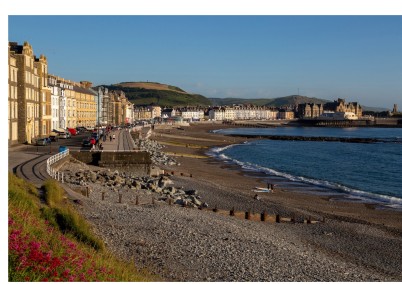

A row of multi-story buildings is to the left of and adjacent to the sea.

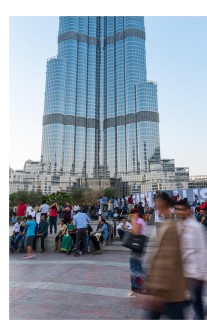

A tall, modern skyscraper with a glass facade is situated above and behind a crowd of people.

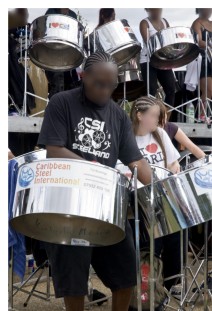

A man with braided hair wearing a black t-shirt is in the midst of a group of steel drums.

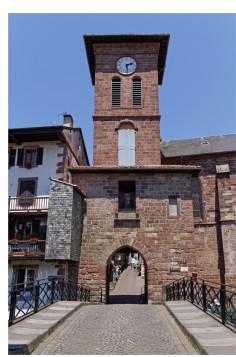

A red brick tower with a clock contains an arched stone gateway.

Figure A2: **Samples from SUBench**.

correcting grammatical errors, refining sentence structure, and enhancing overall readability. All scientific contributions, ideas, and analyses presented are entirely our own. The LLM was not used for research ideation or to retrieve related work.

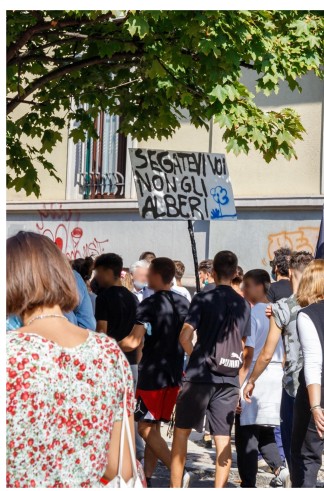

A protest sign with Italian text is in the midst of a group of people.

The model's output has been evaluated against criteria A, B, C, and D. The final score is based on A, B, and C.
A. Object Description Validity: Both object descriptions ('A protest sign with Italian text', 'A group of people') are excellent. They are accurate, clearly detectable, uniquely identifiable within the image context, and adhere to all rules, including correct article usage and avoidance of spatial locators.
B. Spatial Relationship Accuracy & Adherence: The chosen relationship `DR-0` (amid) is factually correct, and the caption 'in the midst of' is a valid and nuanced synonym. The label and caption are consistent, and the single label adheres to the vocabulary rules.
C. Relationship Unambiguity, Strength, & Completeness: This is where the primary failure occurs. While the `amid` (DR-0) relationship is correct and strong, it is critically incomplete. The protest sign is also very clearly and saliently `above` (VDR-0) the group of people. The instructions explicitly state: 'If two relationships from different categories... are both clearly and accurately describing the object pair, you MUST combine them into a compound relationship.' The model failed to do this. The vertical position of the sign is a highly significant spatial dimension, and its omission constitutes a moderate completeness issue. The correct label should have been the compound relationship `VDR-0/DR-0`.
D. `reason` Field Evaluation: The provided `reason` is well-articulated but reveals the model's flawed logic. It correctly identifies the `amid` relationship but then incorrectly dismisses the `above` relationship as 'less salient'. In the context of a protest, a sign being held high is a primary and highly salient feature. This poor judgment directly led to the completeness error identified in Criterion C. This evaluation of the `reason` does not impact the score but highlights the source of the error.

Figure A3: **Filtered Example 1**.

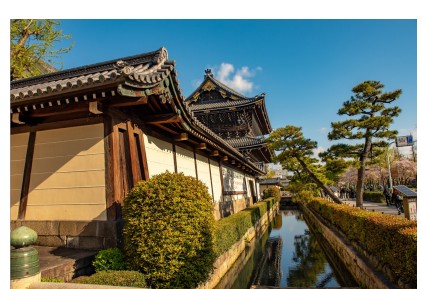

A traditional Japanese building is behind and to the left of a narrow canal.

Overall assessment based on Criteria A, B, C. A. Object Description Validity: The object descriptions ('a traditional Japanese building with a tiled roof', 'a narrow canal') are excellent. They are accurate, clearly detectable, unique within the image context, and correctly use indefinite articles without referencing spatial location.
B. Spatial Relationship Accuracy & Adherence: The output adheres to the label-caption consistency and compound relationship ordering rules (`ADR` before `LDR`). The 'left' (`LDR-0`) component is factually correct from the required Relative (Viewer-Centric) Frame. However, the 'behind' (`ADR-1`) component is a weak and highly debatable choice. The building and the canal are both elongated objects that occupy a similar and overlapping range of depth from the foreground to the mid-ground. While one could argue the canal *starts* closer to the viewer, making the building conceptually 'behind', this is not a strong or clear-cut directional relationship.
C. Relationship Unambiguity, Strength, & Completeness: This is the primary area of failure. The relationship is incomplete and poorly chosen. It omits the most salient, strong, and unambiguous spatial relationship between the two objects: `near` (PDR-0). The building is immediately adjacent to the canal. A far more accurate and complete description would have been 'to the left of and near' (`LDR-0/PDR-0`). By selecting the weak and ambiguous 'behind' relationship over the strong and obvious 'near' relationship, the model has failed to provide the best and most complete description of the scene. This constitutes a moderate completeness issue.
D. `reason` Field Evaluation: The provided `reason` is clear and accurately explains the model's logic for choosing 'behind' (i.e., the canal starts in the foreground). While the logic is understandable, it leads to a suboptimal relationship choice as noted in C. This evaluation does not impact the score.

Figure A4: **Filtered Example 2**.

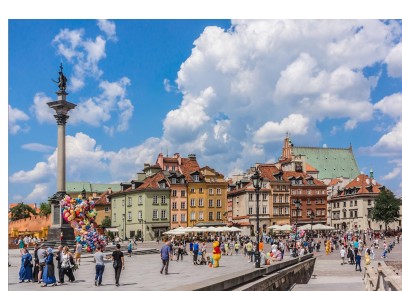

A large group of people is distributed around a tall monument with a statue on top.

Overall assessment based on Criteria A, B, C. A. Object Description Validity: The object descriptions are excellent. 'A large group of people' correctly treats the crowd as a single entity per the grouping rule. 'A tall monument with a statue on top' is accurate, specific, and uniquely identifies the object in the image. Both descriptions correctly use indefinite articles and avoid spatial locators.
B. Spatial Relationship Accuracy & Adherence: This criterion reveals a major flaw. While the `label` (DR-1) and `caption` ('around') are consistent with each other, the relationship itself is factually incorrect based on the visual evidence. The monument is positioned on the far left of the square from the viewer's perspective, while the vast majority of the people are gathered in the center and to the right of it. The people are not 'around' or 'surrounding' the monument in any meaningful way. The model has incorrectly applied a distributional relationship where a simple lateral one is dominant.
C. Relationship Unambiguity, Strength, & Completeness: The chosen relationship is highly ambiguous and weak. The visual evidence for 'around' is minimal to non-existent. In contrast, there is extremely strong and unambiguous evidence for a lateral directional relationship. The relationship is critically incomplete because it completely ignores the most salient spatial fact: that the monument is to the left of the group of people (or conversely, the people are to the right of the monument). A far more accurate and stronger relationship would be `LDR-0` ('left') or `LDR-1` ('right').
D. `reason` Field Evaluation: The `reason` is well-structured and demonstrates an understanding of the different relationship types. However, its core premise that 'the people are scattered in all directions surrounding its base' is a significant misinterpretation of the image. This flawed observation led directly to the selection of the incorrect 'around' relationship. This feedback does not impact t...

Figure A5: **Filtered Example 3**.

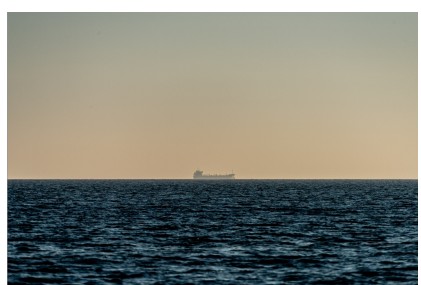

A distant cargo ship is on top of a large body of dark, choppy water.

Overall assessment based on Criteria A, B, C. A. Object Description Validity: The object descriptions ('a distant cargo ship', 'a large body of dark, choppy water') are excellent. They are accurate, clearly detectable, uniquely identifiable within the image, and correctly use indefinite articles without relying on spatial locators.
B. Spatial Relationship Accuracy & Adherence: The chosen relationship `VDR-0` ('above') and its corresponding caption 'on top of' are technically correct from the viewer's 2D Relative Frame, as the ship is at a higher vertical position in the image plane than the water in the foreground. The label and caption are consistent. However, this relationship borders on being a non-salient, generic truth ('a ship is on the water'), which the instructions explicitly warn against.
C. Relationship Unambiguity, Strength, & Completeness: This is the primary area of failure. The output is critically incomplete. While 'above' (VDR-0) is visually true, the model has completely omitted the more significant and salient relationship of 'behind' (ADR-1). The composition strongly emphasizes the depth of the scene, with the vast body of water extending from the viewer into the distance, placing the ship clearly 'behind' the foreground water. The most accurate and complete description would be a compound relationship 'above and behind' (VDR-0/ADR-1). By failing to include the 'behind' dimension, the model has missed the most defining spatial characteristic of the scene.
D. `reason` Field Evaluation: The provided `reason` accurately justifies the choice of `VDR-0` from the relative frame. However, its analysis is superficial as it fails to consider or justify the exclusion of the more powerful 'behind' relationship, indicating a weakness in its comprehensive spatial reasoning. (This observation does not affect the score).

Figure A6: **Filtered Example 4**.

Original Image                                   Negated Image(s)

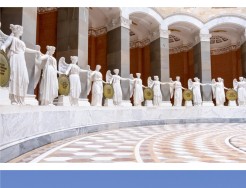
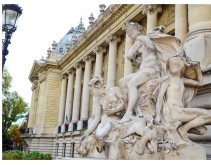
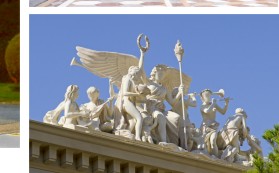
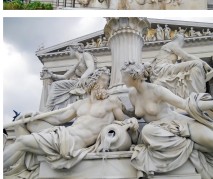

A group of white marble statues depicting classical figures is positioned in front of and far from a large, ornate neoclassical building.

Figure A7: **Negated Example 1**.

Original Image                    Negated Image(s)

A cluster of Tibetan-style buildings is in
the midst of a group of lush green trees.

Figure A8: **Negated Example 2**.

Original Image                    Negated Image(s)

A cluster of Tibetan-style buildings is in
the midst of a group of lush green trees.

Figure A9: **Negated Example 3**.

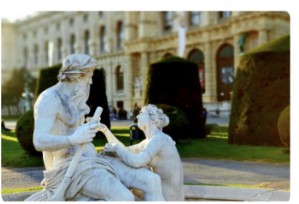

Based on the provided image. Please generate an image with "A group of white marble statues depicting classical figures is positioned behind a large, ornate neoclassical building."

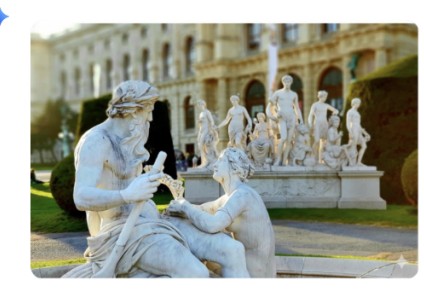

Figure A10: **Case Study #1 of Nano Banana.**.

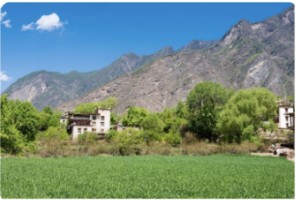

Based on the provided image. Please generate an image with "A cluster of Tibetan-style buildings is out of a group of lush green trees."

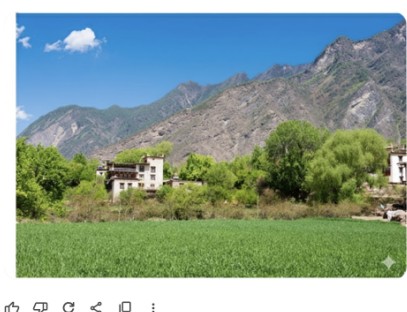

Figure A11: **Case Study #2 of Nano Banana.**.

