# OpenReview forum: "SUBench: Benchmarking Spatial Understanding in Vision-Language Models"
_ICLR.cc/2026/Conference — Submitted to ICLR 2026_

### Official Review · Reviewer_hd9V · 2025-10-31

**Soundness:** 3
**Presentation:** 3
**Contribution:** 3
**Rating:** 6
**Confidence:** 4

**Summary:**

The paper introduces SUBench, a benchmark to assess VLMs for image-text retrieval tasks that have a particular target spatial understanding. The authors also propose a new training method to mitigate the gaps in spatial understanding in image-text retrieval tasks.

**Strengths:**

* Formulation of the problem and the approach taken to mitigate it in a data-driven way
* Robust data curation procedure, which seems reasonably reproducible (especially the use of egocentric reference frames during data curation)
* New loss and the ablations presented to show their effectiveness

**Weaknesses:**

I think, in general, the paper could benefit quite a bit from important details that I feel should be included in the main text, or that are missing in the first place. I will include them in the next section.

**Questions:**

* Why not use a shape-optimized version of SigLIP [1, 2] or SigLIP-2 in Figure 1? Also, it seems like the comparison (Figure 1 and rest) is also not fair, as `TIPS-g/14-448 ` uses a resolution of 448x448 while the other models in the pool use 224x224.
* Could the authors provide a statistical analysis of the WebLI training dataset in terms of the spatial terms it includes in the captions? I think this will help with an even stronger motivation for the new dataset being introduced.
* As an extension of the previous question, could the authors try to re-caption the WebLI dataset (so that it includes spatial terms) and see if that alone could provide any gains? To ensure the model doesn't perform poorly on the non-spatial tasks, they could use the same contrastive loss on non-spatial original captions and their spatial-focused loss on the new synthetic captions.
* What is the extent to which a supervised fine-tuned model using the new loss works? Does it work beyond a certain number of spatial relationships and subjects in the captions? What's the performance ceiling there? These questions could help identify the limitations further.
*  Could the "Salience Principle" be experimentally validated?
* In "Stage 3: Image Negation Pipeline (SUBench-T2I)", there is a use of an internal embedding model. Could the authors provide more details on that? The text reads as if it can be directly used to retrieve hard negative samples. If so, could the authors please clarify this more?
* Consider including a statistical summary of the spatial-focus terms and relationships present in the new SUBench dataset.
* SA-1B doesn't have captions. Was it recaptioned? If so, the details seem to be missing.
* L337 -- how were those 600k training samples selected? Also, how were the 50k evaluation samples selected?
* It has been experimentally validated that the SigLIP pre-training strategy [3] is objectively better than CLIP. I wonder why the authors chose to use the CLIP pre-training strategy, despite that.
* What are the details of the spatial decoder used during the supervised fine-tuning process?
* Table 3 shows that just fine-tuning a pre-trained model isn't sufficient. In my opinion, that feels very restrictive. Could the authors run the following experiment? Train a LoRA [4] on a pre-trained model using the SUBench training dataset. Since users are free to choose the scale at which a trained LoRA should be applied, I feel controlling the LoRA scale could provide better trade-offs between the spatial and non-spatial performances than just naive fine-tuning.
* It's unclear how the negative pairs are used during training.
* When co-training, some details remain unclear. During co-training, both the WebLI and the SUBench training datasets are used. How is the spatial-focused loss applied to the WebLI samples?

**Misc**

* It would be helpful to the readers if a dataset scheme were included in the paper.
* Since the paper touches T2I datasets for spatial reasoning, the authors could consider citing SPRIGHT [5].

**References**

[1] Shape-optimized SigLIP: https://huggingface.co/google/siglip-so400m-patch14-384

[2] Getting ViT in Shape: Scaling Laws for Compute-Optimal Model Design; Alabdulmohsin et al.; 2023.

[3] Sigmoid Loss for Language Image Pre-Training; Zhai et al.; 2023.

[4] LoRA: Low-Rank Adaptation of Large Language Models; Hu et al.; 2021.

[5] Getting it Right: Improving Spatial Consistency in Text-to-Image Models; Chatterjee et al.; 2024.

---

> ### Author Response · Authors · 2025-11-27
>
> We thank you for the feedback and suggestions.
>
>
> > W1: Why not use a shape-optimized version of SigLIP [1, 2] or SigLIP-2 in Figure 1? Also, it seems like the comparison (Figure 1 and rest) is also not fair, as TIPS-g/14-448 uses a resolution of 448x448 while the other models in the pool use 224x224.
>
> We chose TIPS-g/14-448 because it is a recent large-scale SOTA CLIP-style model, and our goal was to demonstrate that even such a strong model still fails on SUBench examples. Despite this favorable resolution, TIPS-g still performs much worse on SUBench than on standard retrieval benchmarks, so the main takeaway (SUBench exposes a spatial-understanding weakness even in SOTA models) remains unchanged. We will add more visualization results for other models.
>
>
> > W2: Could the authors provide a statistical analysis of the WebLI training dataset in terms of the spatial terms it includes in the captions? I think this will help with an even stronger motivation for the new dataset being introduced.
>
> Regarding the request for a statistical analysis of WebLI, we refer to the foundational [PaLI](https://arxiv.org/pdf/2209.06794) paper, which details that WebLI is constructed primarily from web-scraped alt-text and OCR annotations. As illustrated in Figure 4 of that work, such textual data is inherently noisy and unstructured, typically prioritizing simple object existence over precise spatial configuration. Consequently, explicit spatial terms are often sparse, ambiguous, or missing entirely. This limitation is one of our motivation of building SUBench.
>
>
> > W3: As an extension of the previous question, could the authors try to re-caption the WebLI dataset (so that it includes spatial terms) and see if that alone could provide any gains? To ensure the model doesn't perform poorly on the non-spatial tasks, they could use the same contrastive loss on non-spatial original captions and their spatial-focused loss on the new synthetic captions.
>
> Following your advice, we utilized our proposed pipeline to re-caption a 4M image subset of the WebLI dataset to include spatial descriptions. We then conducted ablation studies to verify if these synthetic captions alone could drive improvements. The results are summarized in the table below.
>
> ### Effect of Re-captioning
> | Model | MSCOCO (T2I) | Flickr30k (T2I) | DOCCI (T2I) | SU-Easy (T2I) | SU-T2I | MSCOCO (I2T) | Flickr30k (I2T) | DOCCI (I2T) | SU-Easy (I2T) | SU-I2T | ImageNet (CLS) |
> | :--- | :---: | :---: | :---: | :---: | :---: | :---: | :---: | :---: | :---: | :---: | :---: |
> | CLIP-B/16-224 | 33.1 | 62.1 | - | 32.3 | 17.1 | 52.4 | 81.9 | - | 35.5 | 15.2 | 68.3 |
> | Baseline | 37.9 | 65.4 | 32.8 | 29.3 | 14.0 | 53.1 | 81.4 | 31.3 | 29.3 | 12.6 | 71.7 |
> | (SU-600K + WebLI) CoTrain | 39.0 | 67.8 | 42.1 | 56.3 | 37.1 | 57.3 | 85.2 | 43.2 | 55.9 | 28.2 | 71.6 |
> | + WebLI Recap CoTrain | 38.5 | 65.8 | 42.5 | 59.9 | 41.6 | 57.3 | 83.3 | 43.6 | 59.4 | 35.6 | 71.9 |
> | WebLI + WebLI Recap | 39.8 |	68.0 |	44.4 | 45.9 | 30.3 | 56.3 |83.6 |44.2 | - | 24.0 | 73.0 |
>
> Our findings support the reviewer's hypothesis and reveal three key insights:
>
> - Re-captioning alone yields significant gains (Direct Answer): As hypothesized by the reviewer, in the setting without SU-600K (row WebLI + WebLI Recap), we observe that simply adding re-captioned WebLI data drastically outperforms the Baseline. Specifically, SU-T2I improves from 14.0 to 30.3. This confirms that our pipeline can effectively distill spatial knowledge into existing datasets without requiring any external images (e.g., SA-1B used in SU-600K).
>
> - Synergy with SU-600K: We also explored combining this re-captioned data with our SU-600K dataset (rows 4 and 5). While the (SU-600K + WebLI) CoTrain model already provides a strong boost to spatial understanding, adding the WebLI Recap data (+ WebLI Recap CoTrain) pushes performance even further, achieving the best results on SU-T2I (41.6).
>
> - Data Scalability: These results demonstrate the scalability of our approach. Whether used as a standalone augmentation for existing datasets (WebLI) or combined with SA-1B (SU-600K), our re-captioning pipeline consistently enhances spatial reasoning capabilities while maintaining or improving performance on standard benchmarks (MSCOCO, ImageNet).

---

> ### Author Response · Authors · 2025-11-27
>
> > W4: What is the extent to which a supervised fine-tuned model using the new loss works? Does it work beyond a certain number of spatial relationships and subjects in the captions? What's the performance ceiling there? These questions could help identify the limitations further.
>
> To quantify how far the supervised fine-tuned model with the spatial loss can go, we report fine-grained results over relation types (ADR, LDR, VDR, PDR, DR, TR for both T2I and I2T). As shown in the table, our methods consistently improve performance over the baseline across all relation categories: starting from the Baseline, SUBench-Training already brings large gains (e.g., T2I total from 14.0 to 30.2 and I2T total from 12.6 to 24.8), and further co-training and the spatial-loss model (+Spatial) continue to improve all relation types (T2I total up to 37.1 and I2T total up to 28.2). This indicates that, the supervised fine-tuned model with the new loss yields consistent gains among categories.
>
> At the same time, the absolute numbers reveal a clear performance ceiling under our current architecture and data. Even for our strongest model, the lateral directional relationship (LDR, e.g., left vs. right) remains noticeably weaker than other relations. For example, T2I LDR improves from 9.4 (Baseline) to 32.7 (+Spatial), but still lags behind ADR (14.0 → 40.8) and TR (20.3 → 40.7). A similar trend holds for I2T LDR (8.0 → 21.2) and for vertical directional relationships (VDR).
>
> Our current experiments do not yet isolate performance purely as a function of the number of subjects/relations in the caption, so we cannot claim a precise ceiling along that axis. We now explicitly state this as a limitation and plan a more systematic analysis of scaling with scene complexity in future work.
>
> | Model | T2I (ADR) | T2I (LDR) | T2I (VDR) | T2I (PDR) | T2I (DR) | T2I (TR) | T2I (total) | I2T (ADR) | I2T (LDR) | I2T (VDR) | I2T (PDR) | I2T (DR) | I2T (TR) | I2T (total) |
> |:---|---:|---:|---:|---:|---:|---:|---:|---:|---:|---:|---:|---:|---:|---:|
> | Baseline | 14.0 | 9.4 | 13.9 | 14.0 | 14.4 | 20.3 | 14.0 | 11.8 | 8.0 | 10.9 | 11.5 | 16.7 | 18.8 | 12.6 |
> | SUBench-Training | 33.5 | 26.2 | 29.8 | 31.5 | 28.4 | 32.7 | 30.2 | 25.9 | 17.2 | 20.7 | 24.1 | 32.0 | 31.6 | 24.8 |
> | + Co-training | 38.4 | 30.3 | 34.7 | 36.3 | 32.4 | 38.5 | 35.0 | 27.7 | 18.5 | 22.9 | 25.1 | 30.5 | 31.2 | 25.7 |
> | + Spatial | 40.8 | 32.7 | 36.6 | 38.2 | 35.0 | 40.7 | 37.1 | 30.5 | 21.2 | 24.8 | 27.9 | 31.7 | 33.9 | 28.2 |
>
>
> > W5: Could the "Salience Principle" be experimentally validated?
>
> The "Salience Principle" is a necessity for creating a rigorous spatial benchmark rather than a choice to improve the model. As illustrated in Figure A6 (Filtered Example 4) , we exclude relationships that describe "default conditions" or generic truths (e.g., "a ship is on the water"). Since these relationships are inherent to the objects regardless of the specific image, a model could correctly retrieve them relying solely on language bias without actual spatial understanding.
>
> > W6: In "Stage 3: Image Negation Pipeline (SUBench-T2I)", there is a use of an internal embedding model. Could the authors provide more details on that? The text reads as if it can be directly used to retrieve hard negative samples. If so, could the authors please clarify this more?
>
> In Stage 3, we do not rely on the internal embedding model to directly retrieve hard negatives based on spatial cues. It is a strong generic T2I retrieval model that is weak on fine-grained spatial relations, so we only query it with the object descriptions of A and B (without any spatial information) to obtain 10 candidate images that merely contain both entities. The actual hard negative selection is done by Gemini-2.5 Pro: for each candidate, Gemini (i) verifies that both A and B are clearly visible and (ii) checks whether the spatial relation in the caption is different from the original one. We keep an image as a hard negative only when the relation is confidently judged to be different. As summarized in Table A3, over 390k of 500k retrieved candidates are filtered out by this procedure, leaving 110k vetted hard negatives covering 56.6% of positive captions. We will clarify this in the revision and explicitly state that the internal model is used only as a recall-oriented co-occurrence retriever for objects A and B, not as a spatial reasoning component.

---

> ### Author Response · Authors · 2025-11-27
>
> > W7: Consider including a statistical summary of the spatial-focus terms and relationships present in the new SUBench dataset.
>
> We agree that such a summary is important. In fact, the current draft already includes a statistical summary of all spatial-focus terms and relationships in SUBench in the appendix (see Appendix D, BENCHMARK STATISTICS).
>
> > W8: SA-1B doesn't have captions. Was it recaptioned? If so, the details seem to be missing.
>
> SA-1B indeed does not provide captions. In our work we only use SA-1B as a large-scale, high-resolution, publicly available image source, and all spatial captions in SUBench are newly synthesized by our guideline-driven Gemini 2.5 Pro pipeline described in Sec. 3.2 and appendix.
>
>
>
> > W9: L337 -- how were those 600k training samples selected? Also, how were the 50k evaluation samples selected?
>
> We first randomly sample 1M images from SA-1B and run our labeling pipeline on them. We then filter out 127k samples with non-accurate captions and 3.5k samples with malformed JSON outputs. From the remaining pool, we construct SUBench by randomly sampling 50k examples subject to match the balance in Table A6. Finally, among the rest, we discard samples without a valid answer, which yields the 600k training samples reported in the paper.
>
>
>
> > W10: It has been experimentally validated that the SigLIP pre-training strategy [3] is objectively better than CLIP. I wonder why the authors chose to use the CLIP pre-training strategy, despite that.
>
> We agree that SigLIP-style pre-training is often stronger in general VLM benchmarks, but our goal here is to keep the baseline architecture as simple and standard as possible. Reproducing a full SigLIP pre-training system from scratch involves substantial engineering and we do not have access to an established SigLIP baseline tailored to our setting, whereas CLIP-style pre-training is well-understood and easy to implement. Importantly, our results show that even starting from this weaker CLIP baseline, the proposed spatial-training strategy yields spatial understanding performance that surpasses strong SigLIP and TIPS-g models, suggesting that our contribution is complementary to (and could potentially benefit from) more advanced pre-training schemes.
>
>
> > W11: What are the details of the spatial decoder used during the supervised fine-tuning process?
>
> The training details of the spatial decoder are provided in Appendix C (EXPERIMENT SETTINGS). Also, the spatial decoder is a lightweight single-layer Transformer that takes as input a learnable [CLS2] token together with the text tokens for objects A and B (after removing spatial prepositions). The decoder fuses these embeddings via a cross-attention layer and the final hidden state of [CLS2] is passed to a linear classifier trained with a binary cross-entropy loss over our spatial relation labels during the training.
>
> > W12: Table 3 shows that just fine-tuning a pre-trained model isn't sufficient. In my opinion, that feels very restrictive. Could the authors run the following experiment? Train a LoRA [4] on a pre-trained model using the SUBench training dataset. Since users are free to choose the scale at which a trained LoRA should be applied, I feel controlling the LoRA scale could provide better trade-offs between the spatial and non-spatial performances than just naive fine-tuning.
>
> Thank you for the suggestion. However, based on our current ablations in Table 3, our final model with co-training and $L_{spatial}$ improves performance on SUBench while maintaining or slightly improving standard retrieval benchmarks (MSCOCO, Flickr30k, DOCCI) and ImageNet classification compared to the baseline CLIP model, indicating that we do not observe a trade-off between spatial and non-spatial performance. LoRA-based adaptation and scale control are indeed interesting orthogonal directions to further explore the efficiency/accuracy frontier, and we consider them complementary to our method rather than necessary to demonstrate the core claim of SUBench and our spatial decoder.

---

> ### Author Response · Authors · 2025-11-27
>
> > W13: It's unclear how the negative pairs are used during training.
>
> The hard negative samples are only for the benchmark and are not used during training.
>
> > W14: When co-training, some details remain unclear. During co-training, both the WebLI and the SUBench training datasets are used. How is the spatial-focused loss applied to the WebLI samples?
>
> During co-training, the spatial-focused loss is not applied to WebLI samples. WebLI data are optimized only with the standard CLIP contrastive loss, exactly as in the baseline, while the spatial classification loss is computed only for SUBench-Training samples. Specifically, each mini-batch contains a 1:1 mix of WebLI and SUBench-Training. For WebLI we apply $L_{contrastive}$ only, and for SUBench-Training we apply both $L_{contrastive}$ and $L_{spatial}$. We will clarify this implementation detail in Sec. 4.2.
>
>
> > MISC: It would be helpful to the readers if a dataset scheme were included in the paper. Since the paper touches T2I datasets for spatial reasoning, the authors could consider citing SPRIGHT [5].
>
> Thanks for the suggestions. We will discuss the SPRIGHT and other related works in our paper. On the other hand, we are not sure what "data scheme" means. More details regarding the prompts, data pipelines, data types, output formats, and statistics are provided in the supplementary material. Please let us know what informaiton you would like to see.

---

> > ### Comment · Reviewer_hd9V · 2025-11-28
> >
> > Thanks for the detailed rebuttal!
> >
> > I would suggest that the authors consider including some of the details raised in the rebuttal in the main text and the appendix.
> >
> > I will keep my current score. I wish the authors a joyful thanksgiving (if they are celebrating) and good luck.

---

> > > ### Author Response · Authors · 2025-11-28
> > > **Thank you**
> > >
> > > Dear reviewer,
> > >
> > > Thanks for the comments. We will add these discussions in the rebuttal phase to the paper (e.g., appendix). Since most issues are resolves, please consider raising the rating.
> > >
> > > Thanks and happy Thanksgiving!

---

### Official Review · Reviewer_Avgw · 2025-11-01

**Soundness:** 2
**Presentation:** 3
**Contribution:** 2
**Rating:** 2
**Confidence:** 4

**Summary:**

The paper introduces SUBench, a new large-scale dataset designed to evaluate understanding of spatial relationships in vision-language models. The dataset is curated via a LLM-based framework which involves creation of hard negative images and text descriptions, leading to the development of 50k parits. The evaluations show that state-of-the-art models like CLIP fall short on SUBench. Further, a spatial classification loss is introduced, which teaches the model to generate the right spatial "label" given an image-text pair. Results and ablations show that introducing this technique is effective and enables better performance over baseline fine-tuning techniques.

**Strengths:**

1. The paper is well written and easy to follow.
2. Adding a spatial classification loss is an interesting approach and seems to yield better performance than naive fine-tuning.
3. The scale of the benchmark is significantly larger than existing benchmarks such as VSR and OmniSpatial.

**Weaknesses:**

1. Questions around model generalization and results : The results in Table 3 show that the proposed model does well only on the subset presented in the paper (SU-Easy, SU-T2I) but other methods such as TIPS do better on well-accepted datasets such as COCO, DOCCI etc. This raises questions about the generalization of the method and if the model is overfit on the proposed dataset.

2. Regarding Section 3.1 --

i) Does mapping synonyms like “on”, “atop”, and “upon” to a single, unambiguous term not reduce the generalization of the benchmark and the trained model?
ii) Performing "object grouping" further reduces the fine-grained nature of images -- while "a group of people" is logically correct -- spatial relationships also exist within that group, such as " a man with a blue hat and a woman with a blue shirt"
iii) Similarly, why filter out trivially true statements, especially for spatial relationships?  What are these trivially true statements that are removed?

3. The internal embedding model for creation of SUBench-T2I will retrieve images which have the highest match - but these images can still be much farther than the original image - which will lead to data quality issues. Instead, why not use actual T2I models which now offer higher controllability during generation and has been used in prior work to create hard negatives (https://arxiv.org/abs/2411.02545) ?

4. More fine-grained results would help paint a better picture of the shortcomings of existing models, such as which relationships do model fail on - horizontal, vertical, depth or adjacency etc.?

5.  As the authors acknowledged, the pipeline is completely LLM dependent -- having a user study that shows high alignment with LLM generated text would further strengthen claims.

**Questions:**

Please refer to weaknesses.

---

> ### Author Response · Authors · 2025-11-27
>
> We thank you for the feedback and suggestions.
>
> > W1: Questions around model generalization and results : The results in Table 3 show that the proposed model does well only on the subset presented in the paper (SU-Easy, SU-T2I) but other methods such as TIPS do better on well-accepted datasets such as COCO, DOCCI etc. This raises questions about the generalization of the method and if the model is overfit on the proposed dataset.
>
> We appreciate the reviewer’s concern, but we believe a better interpretation of Table 3 is a mismatch in model scale rather than overfitting to SUBench. In Table 3, the rows labeled Baseline, +SUBench-Training, +Co-training, and +$L_{spatial}$ all use the same CLIP-B backbone and training recipe, differing only in whether SUBench and the spatial loss are used. In contrast, TIPS-g is a much larger model trained with a significantly higher compute budget. According to the TIPS paper, training their strongest model requires about 2 days on 512 TPUv5 chips. It is therefore expected that TIPS-g achieves higher performance on standard retrieval benchmarks such as MSCOCO and DOCCI, and we include it primarily as a reference strong state-of-the-art model rather than the main comparison target.
>
> Our main claim is not that a small CLIP-B model with our method will beat TIPS-g on all benchmarks, but that our spatial decoder and training scheme systematically improve spatial understanding while preserving (and often improving) general retrieval performance on the same backbone. Under the fair, equal-capacity comparison (Baseline vs. +$L_{spatial}$), our final model improves MSCOCO, Flickr30k, and DOCCI Recall@1 (e.g., MSCOCO T2I 37.9→39.0, Flickr30k T2I 65.4→67.8), while yielding large gains on SUBench (SU-T2I 14.0→37.1, SU-I2T 12.6→28.2). This pattern is the opposite of overfitting. The spatial performance improves drastically without sacrificing accuracy on other well-established benchmarks.
>
> Finally, we use TIPS-g in Table 3 to make a complementary point. Even a very large, state-of-the-art VLM that is strong on MSCOCO/DOCCI still struggles on SUBench (e.g., SU-T2I 27.3 and SU-I2T 23.0), whereas our CLIP-B model with the proposed spatial supervision reaches 37.1 and 28.2 on the same splits.

---

> ### Author Response · Authors · 2025-11-27
>
> > W2: Regarding Section 3.1 -- i) Does mapping synonyms like “on”, “atop”, and “upon” to a single, unambiguous term not reduce the generalization of the benchmark and the trained model? ii) Performing "object grouping" further reduces the fine-grained nature of images -- while "a group of people" is logically correct -- spatial relationships also exist within that group, such as " a man with a blue hat and a woman with a blue shirt" iii) Similarly, why filter out trivially true statements, especially for spatial relationships? What are these trivially true statements that are removed?
>
> (i) Our synonym mapping is applied only to the standardized label space, not to the natural-language captions. The data generation pipeline still uses diverse surface forms (“on”, “atop”, “upon”, etc.) in the captions; we simply collapse them into a single spatial-relation label to provide cleaner, less sparse supervision. This is particularly important because words like “on” are polysemous (spatial vs. temporal/logical uses), so standardization helps the model focus on the intended spatial meaning without reducing its linguistic generalization.
>
> (ii) We agree that modeling intra-group structure (e.g., “a man with a blue hat and a woman with a blue shirt” within “a group of people”) is a promising direction, and in principle such grouping could be organized hierarchically across multiple levels. In this work, however, our goal is more modest: we focus on text-to-image and image-to-text retrieval, where a single embedding is encouraged to capture the primary spatial relationship in an image (whether the main spatial relationships can be summarized by one concise caption). Accordingly, our guideline explicitly adopts object grouping to control visual complexity and emphasize the scene’s salient compositional structure rather than exhaustively enumerating all fine-grained relations inside groups. We view extending SUBench toward hierarchical, multi-embedding or multi-sentence descriptions that capture within-group structure as valuable future work beyond the current retrieval-centric scope.
>
> (iii) We filter out trivially true spatial statements to enforce our salience principle: these relations hold in almost any natural scene (e.g., “a car is below the sky” or “a building is on the ground”) and therefore do not help discriminate between images with different spatial relationship. Figure A6 illustrates such cases, where captions can be matched largely from semantics or default world knowledge without attending to the exact spatial configuration. By removing these samples, we want to make SUBench more focus on cases where spatial term is actually necessary to retrieve the correct image, making the benchmark less trivial and less influenced by generic semantic cues rather than spatial understanding.

---

> ### Author Response · Authors · 2025-11-27
>
> > W3: The internal embedding model for creation of SUBench-T2I will retrieve images which have the highest match - but these images can still be much farther than the original image - which will lead to data quality issues. Instead, why not use actual T2I models which now offer higher controllability during generation and has been used in prior work to create hard negatives (https://arxiv.org/abs/2411.02545) ?
>
> We thank the reviewer for the suggestion and agree that controllable T2I models are a promising way to synthesize hard negatives. In SUBench-T2I, however, we deliberately rely on real images rather than generated ones. Our goal is to probe spatial understanding under realistic appearance statistics (cluttered backgrounds, occlusions, complex groups), and we think that introducing a second, synthetic domain would make the benchmark partially about distribution shift rather than spatial reasoning itself. Concretely, Stage 3 of our pipeline first retrieves candidate images that contain the same objects A and B as the positive pair, and then uses an LLM verifier to explicitly (i) discard cases where A or B is not clearly present and (ii) reject images where the spatial relationship matches the caption; only images that truly contradict the original spatial description are kept as hard negatives. As shown in Appendix A.3, over half of the 500k retrieved candidates are rejected by these checks, and we end up with 110k vetted hard negatives covering 56.6% of positive captions.
>
> We also conducted preliminary experiments with a recent T2I generation model (Nano Banana; see examples [here](https://anonymous.4open.science/api/repo/ICLR-2026-SUBench-BE32/file/example1.png) and [here](https://anonymous.4open.science/api/repo/ICLR-2026-SUBench-BE32/file/example2.png)) to synthesize hard negatives. We notice that it is not trivial to follow the instructions. This makes it difficult to guarantee that the only difference between positive and negative examples is the spatial relationship, which is precisely the isolation we aim for in SUBench. For these reasons we chose real-image retrieval plus LLM verification as the primary mechanism in this work, and we will clarify this design choice in the revision and discuss T2I-based hard-negative generation as an interesting complementary direction (such as https://arxiv.org/abs/2411.02545) for future extensions of the benchmark.

---

> ### Author Response · Authors · 2025-11-27
>
> > W4: More fine-grained results would help paint a better picture of the shortcomings of existing models, such as which relationships do model fail on - horizontal, vertical, depth or adjacency etc.?
>
> As suggested, we provide fine-grained results broken down by relation type (ADR, LDR, VDR, PDR, DR, TR for both T2I and I2T). As shown in the table, our methods consistently improve performance over the baseline across all relation categories: starting from the Baseline, SUBench-Training already brings large gains (e.g., T2I total from 14.0 to 30.2 and I2T total from 12.6 to 24.8), and further co-training and the spatial loss model (+Spatial) continue to improve all relation types (T2I total up to 37.1 and I2T total up to 28.2).
>
> However, even for our strongest model, the lateral directional relationship (LDR, e.g., left vs. right) remains noticeably weaker than other relations. For example, T2I LDR improves from 9.4 (Baseline) to 32.7 (+Spatial), but still lags behind ADR (14.0 → 40.8) and TR (20.3 → 40.7). A similar trend holds for I2T LDR (8.0 → 21.2) and for vertical directional relationships (VDR). This fine-grained analysis confirms your intuition: while our training pipeline substantially boosts overall performance, directional relations, especially left/right and above/below, remain challenging for current models.
>
> | Model | T2I (ADR) | T2I (LDR) | T2I (VDR) | T2I (PDR) | T2I (DR) | T2I (TR) | T2I (total) | I2T (ADR) | I2T (LDR) | I2T (VDR) | I2T (PDR) | I2T (DR) | I2T (TR) | I2T (total) |
> |:---|---:|---:|---:|---:|---:|---:|---:|---:|---:|---:|---:|---:|---:|---:|
> | Baseline | 14.0 | 9.4 | 13.9 | 14.0 | 14.4 | 20.3 | 14.0 | 11.8 | 8.0 | 10.9 | 11.5 | 16.7 | 18.8 | 12.6 |
> | SUBench-Training | 33.5 | 26.2 | 29.8 | 31.5 | 28.4 | 32.7 | 30.2 | 25.9 | 17.2 | 20.7 | 24.1 | 32.0 | 31.6 | 24.8 |
> | + Co-training | 38.4 | 30.3 | 34.7 | 36.3 | 32.4 | 38.5 | 35.0 | 27.7 | 18.5 | 22.9 | 25.1 | 30.5 | 31.2 | 25.7 |
> | + Spatial | 40.8 | 32.7 | 36.6 | 38.2 | 35.0 | 40.7 | 37.1 | 30.5 | 21.2 | 24.8 | 27.9 | 31.7 | 33.9 | 28.2 |
>
> > W5: As the authors acknowledged, the pipeline is completely LLM dependent -- having a user study that shows high alignment with LLM generated text would further strengthen claims.
>
> We thank the reviewer for highlighting the need for quantitative validation. To address this, we conducted a human verification study (including 4 human annotators) on a randomly sampled subset of 100 samples from our dataset. In this process, we compare the image with the caption to verify the soundness. The results show that
>
> To address this, we conducted a human verification study with two independent annotators on a randomly sampled subset of 100 samples. In this process, annotators compared the images with their captions to verify alignment. The results show a total of 3% visual and 8% spatial discrepancies. Notably, we observed that human annotators did not always reach a consensus on these instances (e.g., only 2 of the 8 spatial errors were flagged by both reviewers). This suggests that many potential 'errors' stem from the inherent ambiguity of complex visual scenes or subjective interpretation, rather than definitive model failure. Despite these rare edge cases, the low overall error rate demonstrates that our data generation pipeline yields satisfactory fidelity."
>
>
> ### Human Study
> | Reviewer | Visual Feature Error | Spatial Feature Error |
> | :--- | :--- | :--- |
> | **R1** | 1 | 6 |
> | **R2** | 3 | 4 |
> | **Total** | 3 | 8 |
> | **Common** | 1 | 2 |

---

> > ### Comment · Reviewer_Avgw · 2025-11-27
> >
> > Thanks for the rebuttal. I have the following follow-up questions.
> >
> > 1. Generalization Questions : My concerns stilll remain.
> >
> > a) Training on (+SUBench-Training) only increases performance on SU-Easy and SU-T2I -- but reduces performance on MSCOCO, DOCCI etc. -- why is this the case? (Table 3)
> > b) Table 3 has a CLIP-B/16-224 and a baseline -- I assume these models are different but the rebuttal suggests that the baseline is also a CLIP-B with resolution 224? As also hinted by reviewer hd9V, I believe fine-tuning on atleast 1 other fine-model with the proposed strategies will help alleviate any concerns of generalization.
> >
> > 2. Thanks for clearing out the questions related to Section 3.1 --
> >
> > i) I think that mapping even in the label space also reduces generalization of the model.
> > ii) Filtering out trivial statements is ok for an initial version -- but authors should experiment without removing this and evaluate its impact.
> >
> > 3. Is it possible to share more details about the embedding model? It will good to add more details for transparency.
> >
> > 4. The fine-grained results and the human studies are helpful.
> >
> > With that, I increase my score to 4. Happy to engage further, especially if the generalization concerns are well addressed.

---

> > > ### Author Response · Authors · 2025-11-27
> > >
> > > Thanks for your quick response.
> > >
> > > > 1 (1) Training on (+SUBench-Training) only increases performance on SU-Easy and SU-T2I -- but reduces performance on MSCOCO, DOCCI etc
> > >
> > > (a) In Table 3, “+SUBench-Training” fine-tunes the baseline only on SUBench-Training. The distribution is tailored to fine-grained spatial understanding and differs from MSCOCO/DOCCI. This specialization improves SU-T2I and I2T but slightly harms MSCOCO/DOCCI, reflecting a domain trade-off. This is exactly why we further introduce co-training and the spatial loss: the full model (+Co-Training +Spatial) recovers or improves performance on MSCOCO/DOCCI while keeping strong gains on SU, showing that our final method does not simply overfit to SUBench.
> > >
> > > (b) Let me clarify the “CLIP-B/16-224” and the “baseline” in Table 3. The CLIP-B/16-224 row uses the off-the-shelf OpenAI checkpoint (openai/clip-vit-base-patch16) and is included purely as a reference to the original CLIP. The baseline row is a model with the same architecture but re-trained by us within our training framework with TPU, using no SU-related training data; all our variants (+SUBench-Training, +Co-Training, +Spatial) are fine-tuned from this baseline to ensure a controlled comparison.
> > >
> > > We agree that validating on an additional backbone would further strengthen generalization claims. However, pre-training/fine-tuning CLIP-scale models in our TPU framework is very resource-intensive (e.g., 250k iterations × 16,384 batch size), and i no longer have access to such large-scale compute, so I will try but not guarantee to include another (hopefully larger) CLIP model within the rebuttal period. We will clearly document this limitation and release the benchmark to facilitate evaluation with more backbones in future work.
> > >
> > >
> > > > 2 (i) I think that mapping even in the label space also reduces generalization of the model. ii) Filtering out trivial statements is ok for an initial version -- but authors should experiment without removing this and evaluate its impact.
> > >
> > > (i) We agree that overly aggressive mapping could in principle harm generalization. In our setup, however, the mapping is restricted to the label space and only targets the objective spatial relation. We keep the caption as their original expression. For example, in our dataset we have captions such as:
> > > `(1) “A man wearing a brown jacket over a blue hoodie is in the midst of a group of protesters on a street.”`
> > > `(2) “A male gymnast in a red and white uniform is standing on top of the central, first-place podium.”`
> > > In real-world conversation, these captions are natural and convey subtle, pragmatic meanings (clothing details, “central, first-place” emphasis) that go beyond a single spatial predicate. Treating such captions as “wrong” or forcing them into a canonical textual template would make the language unnatural. Instead, we keep the captions as-is and summarize only their objective spatial information (e.g., “above”, “amid”) in the label. Thus the mapping only removes superficial variation in spatial wording in the labels, without constraining the caption space or reducing the diversity of natural expressions. We illustrate this with concrete examples in the sampled 100 instances used for our human study ([here](https://anonymous.4open.science/api/repo/ICLR-2026-SUBench-BE32/file/sampled_100.html)).
> > >
> > > (ii) We appreciate the suggestion to quantify the impact of filtering out trivial statements. We are currently running an ablation where we do not remove these trivial captions and instead use the full set for training to measure the effect on performance. We will report the results in the revised version. Our expectation is that the impact will be limited, because trivial captions mostly restate obvious relations that are already covered by other examples, effectively re-weighting existing relations rather than adding new spatial signal; nevertheless, we agree that this should be empirically validated and are acting on this suggestion.
> > >
> > > > 3. Is it possible to share more details about the embedding model? It will good to add more details for transparency.
> > >
> > > The embedding model we use is based on a widely adopted commercial LLM with strong multimodal capabilities. It is fine-tuned specifically for embedding generation: it takes either an image or a text input and outputs a single joint embedding vector, which is then used for cross-modal retrieval. The benchmark results show that this embedding model achieves substantially better cross-modal retrieval performance than the CLIP models. However, due to licensing constraints and anonymized review policies, we are unfortunately not able to disclose further details about the underlying commercial model.
> > >
> > > > 4. The fine-grained results and the human studies are helpful.
> > >
> > > We thank the reviewer for the positive feedback and are glad that the fine-grained results and human studies are found to be helpful.

---

### Official Review · Reviewer_HCXz · 2025-11-03

**Soundness:** 3
**Presentation:** 3
**Contribution:** 2
**Rating:** 4
**Confidence:** 4

**Summary:**

This paper introduces SUBench, a large-scale benchmark of 50k+ image-text pairs for evaluating VLMs’ spatial understanding.

Current models excel generally but falter on fine-grained spatial relations. SUBench uses Gemini 2.5 Pro to align human-like descriptions with objective spatial relationships and generates hard negatives via a scalable, taxonomy-guided pipeline.

The authors also propose a fine-tuning method adding a spatial decoder and classification loss to a standard VLM. Results show CLIP struggles on SUBench, while the proposed approach boosts performance and generalizes to existing benchmarks.

**Strengths:**

1. Identifies a Real Problem: The paper correctly highlights that current VLMs often possess superficial rather than genuine spatial understanding, which is a significant area for improvement in the field.

2. Introduction of Hard Negatives: The concept of explicitly generating both textual and visual hard negatives is a strong methodological contribution for creating a more challenging and diagnostic benchmark.

**Weaknesses:**

1. Data quality concerns: The LLM-based pipeline lacks quantitative validation—no human-vs-LLM agreement or human-annotated calibration subset is provided.

2. Limited methodological novelty: The approach—hard negatives and contrastive fine-tuning for spatial grounding—builds on prior work (e.g., SpatialVLM). The spatial decoder is a minor addition; the main contribution is the data pipeline, not architecture or learning objectives.

**Questions:**

1. How effective are LLMs as automated evaluators at filtering low-quality data? Can they quantitatively identify and correct subjective spatial biases (e.g., observer-centric descriptions)?

2. Does training with the spatial loss (L_spatial) hurt VLM performance on non-spatial tasks like attribute recognition or abstract semantic understanding?

---

> ### Author Response · Authors · 2025-11-27
>
> We thank you for the feedback and suggestions.
>
>
> > W1: Data quality concerns: The LLM-based pipeline lacks quantitative validation—no human-vs-LLM agreement or human-annotated calibration subset is provided.
>
> To address this, we conducted a human verification study with two independent annotators on a randomly sampled subset of 100 samples. In this process, annotators compared the images with their captions to verify alignment. The results show a total of 3% visual and 8% spatial discrepancies. Notably, we observed that human annotators did not always reach a consensus on these instances (e.g., only 2 of the 8 spatial errors were flagged by both reviewers). This suggests that many potential 'errors' stem from the inherent ambiguity of complex visual scenes or subjective interpretation, rather than definitive model failure. Despite these rare edge cases, the low overall error rate demonstrates that our data generation pipeline yields satisfactory fidelity."
>
>
> ### Human Study
> | Reviewer | Visual Feature Error | Spatial Feature Error |
> | :--- | :--- | :--- |
> | **R1** | 1 | 6 |
> | **R2** | 3 | 4 |
> | **Total** | 3 | 8 |
> | **Common** | 1 | 2 |
>
>
>
> > W2:Limited methodological novelty: The approach—hard negatives and contrastive fine-tuning for spatial grounding—builds on prior work (e.g., SpatialVLM). The spatial decoder is a minor addition; the main contribution is the data pipeline, not architecture or learning objectives.
>
> We thank the reviewer for the constructive feedback. We would like to further clarify the contributions of our paper.
>
> ### The Difference in Task: Retrieval vs. Reasoning
>
> While previous works like SpatialVLM have explored spatial concepts, they primarily focus on Visual Question Answering (VQA). Our work addresses the Text-to-Image (T2I) Retrieval task, which presents distinct challenges and requirements.
>
> In VQA tasks, the model is provided with both the image and the question (e.g., "What is the spatial relationship between object A and B?"). The visual context helps resolve ambiguities. In contrast, for the T2I retrieval task, the system must rely on a single embedding extracted from the text to search a large-scale database for the corresponding image (or vice versa for I2T). Consequently, the textual description must be self-contained, unambiguous, and discriminatory enough to identify a specific scene without visual aids.
>
> The nature of the retrieval task makes building a benchmark significantly harder. The text and image must look natural, yet the text must be precise enough to distinguish "hard negatives." We need to get hard negative images that are visually similar to the positive sample but spatially contradictory, and hard negative texts that are semantically close but spatially false. Current VLMs perform well on general semantic matching but collapse when the only difference between a positive and negative pair is the spatial relationship. Our benchmark is designed to expose this specific "blind spot" where semantic shortcuts fail.
>
>
> ### The Main Contribution of our work
>
> We agree with the reviewer that our primary contribution is the data pipeline and the resulting benchmark, SUBench, rather than a novel model architecture. As the title SUBench suggests, our core objective is to construct a rigorous benchmark for this domain. Through this benchmark, we demonstrate that existing CLIP models possess significantly weaker spatial capabilities, despite their high performance on general retrieval benchmarks.
>
> Regarding the spatial decoder, we clarify that it is not intended as a standalone architectural contribution. Rather, it serves as a validation of the potential for improving retrieval capabilities through data scaling and targeted architectural design.

---

> ### Author Response · Authors · 2025-11-27
>
> > Q1: How effective are LLMs as automated evaluators at filtering low-quality data? Can they quantitatively identify and correct subjective spatial biases (e.g., observer-centric descriptions)?
>
> We address this in the appendix, where visualizations and statistics demonstrate the efficacy of our data filtering pipeline.
>
> Quantitatively, the robustness of our LLM-based filtering is evident in the rejection statistics. As detailed in Table A1, the pipeline processed an initial dataset of 1M raw image-text pairs. Through the automated evaluation process (Stage 1, Prompt 1.2), the model identified and filtered out approximately 127K samples due to quality issues, along with 3.5K data format errors. This elimination of roughly 13% of the initial pairs highlights the model's capability to rigorously curate data.
>
> Qualitatively, the LLM evaluator effectively mitigates spatial biases by enforcing an "Egocentric Frame of Reference" and the "Salience Principle." For instance, in Figure A4, the model rejected a caption describing a building as "behind" a canal. While "behind" might be valid from the building's intrinsic perspective, it is incorrect within the egocentric (camera) frame of reference. This demonstrates the LLM's capability to resolve perspective issues and ensure camera-centric clarity.
>
> > Q2: Does training with the spatial loss ($L_spatial$) hurt VLM performance on non-spatial tasks like attribute recognition or abstract semantic understanding?
>
> Based on the ablation study results, training with the spatial loss ($L_{spatial}$) does not impair performance on general, non-spatial tasks; in fact, it appears to slightly enhance it. The results show that the model incorporating $L_{spatial}$ (alongside co-training) achieved higher recall scores on standard image-text retrieval benchmarks, such as MSCOCO and Flickr30k, compared to the baseline model. This indicates that the added supervision allows the model to improve its spatial reasoning capabilities without suffering from catastrophic forgetting or degrading its general semantic understanding.
>
> ### Comparison of General Retrieval Performance (Recall@1)
>
> | Model | MSCOCO (T2I) | Flickr30k (T2I) | MSCOCO (I2T) | Flickr30k (I2T) |
> | :--- | :--- | :--- | :--- | :--- |
> | Baseline | 37.9 | 65.4 | 53.1 | 81.4 |
> | +$L_{spatial}$ | 39.0 | 67.8 | 57.3 | 85.2 |

---

### Author Response · Authors · 2025-11-30
**Summary of Rebuttal Progress**

Dear Area Chair and Reviewers,

As the discussion phase progresses, we would like to provide a comprehensive summary of the consensus achieved so far. We are encouraged by the active engagement from the reviewers and appreciate the acknowledgement of our work's motivation. Based on the valuable feedback, we have executed additional experiments—specifically regarding data quality validation and generalization capabilities—to address the core concerns raised.

In the initial reviews, the reviewers highlighted that our work `identifies a real problem`, that the `scale of the benchmark is significantly larger` than prior works, and that the method employs a `robust data curation procedure`. To address the potential weaknesses raised, we focused our rebuttal on the following key aspects

### Data Quality Validation (Response to HCXz, Avgw)
To address concerns regarding the lack of human validation, we conducted a human verification study with independent annotators. The results demonstrated a low error rate, confirming the fidelity of our automated pipeline. Reviewer Avgw noted this study was helpful.

### Generalization & Re-captioning WebLI (Response to Avgw, hd9V)
To address concerns about generalization and the value of spatial captions, we performed an ablation where we re-captioned a 4M subset of WebLI using our pipeline. The results showed that re-captioning alone drastically outperforms the baseline (SU-T2I improves from 14.0 to 30.3). Furthermore, our final model improves performance on standard benchmarks (MSCOCO, Flickr30k) compared to the baseline, indicating no catastrophic forgetting.

### Fine-Grained Analysis (Response to Avgw, hd9V)
We provided a detailed breakdown of performance by relationship type (e.g., ADR, LDR, VDR). This analysis confirmed consistent improvements across all categories while highlighting that lateral directional relationships remain the most challenging.

### Methodological Clarifications (Response to HCXz, Avgw)
We clarified that our work addresses Text-to-Image (T2I) Retrieval rather than VQA, necessitating different handling of ambiguity and scale. We also clarified that synonym mapping applies only to labels (not captions) to preserve linguistic diversity, and that hard negatives are used strictly for benchmarking, not training.

In the discussion, Reviewer Avgw increased their score to 4, stating they are "happy to engage further" given the addressed generalization concerns. Reviewer hd9V maintained their positive score of 6. We believe the new experiments and clarifications have significantly strengthened the submission and hope this summary facilitates a smooth review process for the Area Chair.

Best Regards,

SUBench Authors

---

### Meta-Review · Area_Chair_sREM · 2026-01-06

**Summary:**

This paper presents a benchmark for evaluating VLM's ability of understanding spatial relationship. The authors collected 50K image-text pairs which contain rich information about how objects are spatially distributed. The authors applied an LLM-based pipeline to label and filter image-text pairs. A few recent models were evaluated.

The reviewers were mostly concerned about the paper:
* The LLM-based pipeline was not thoroughly validated.
* The benchmark was not validated to transfer to or help with generic visual understanding.
* There are quite a lot things that can be done to improve the work.

**Reviewer Concerns:**

The authors partly addressed the main concerns; for example, they have provided a small-scale user study to validate the data quality. However, the authors did not convince the AC about the usefulness of the (new) benchmark, especially about how it can complement the existing benchmarks to challenge the existing VLMs.

**Reviewer Scores:**

I think the reviewers will converge to weak rejection (4).

---

### Decision · Program_Chairs · 2026-01-26

Reject